# Coreset Selection via LLM-based Concept Bottlenecks

## Abstract

Coreset Selection (CS) aims to identify a subset of the training dataset that achieves model performance comparable to using the entire dataset. Many state-of-the-art CS methods select coresets using scores whose computation requires training the downstream model on the entire dataset first and recording changes in the model's behavior on samples as it trains (training dynamics). These scores are inefficient to compute and hard to interpret, as they do not indicate whether a sample is difficult to learn in general or only for a specific downstream model. Our work addresses these challenges by proposing a score that computes a sample's difficulty using human-understandable textual attributes (concepts) independent of any downstream model. Specifically, we measure the alignment between a sample's visual features and concept bottlenecks, derived via large language models, by training a linear concept bottleneck layer and computing the sample's difficulty score using it. We then use stratified sampling based on this score to generate a coreset of the dataset. Crucially, our score is efficiently computable without training the downstream model on the full dataset even once, leads to high-performing coresets for various downstream models, and is computable even for an unlabeled dataset. Through experiments on five diverse datasets including ImageNet-1K, we show that our coresets outperform random subsets, even at high pruning rates, and lead to model performance comparable to or better than coresets found by training dynamics-based methods.

## 1 Introduction

Machine learning (ML) pipelines are increasingly demanding more data and compute (Touvron et al., 2023; Achiam et al., 2023) to achieve improved performance on various tasks. While in line with empirical neural scaling laws (Kaplan et al., 2020; Hestness et al., 2017; Henighan et al., 2020; Rosenfeld et al., 2019) where a model's performance improves with increasing model and training data size, these improvements come at an unsustainable cost of compute/energy. However, recently (Sorscher et al., 2022; Li et al., 2024) demonstrated that data pruning plays a crucial role in enabling an exponential reduction in test error with increasing dataset size, underscoring the importance of data quality over quantity.

Coreset Selection (CS) (Mirzasoleiman et al., 2020; Guo et al., 2022; Paul et al., 2021; Xia et al., 2022; Maharana et al., 2023; Zheng et al., 2022; Choi et al., 2024) is a data pruning technique that improves the efficiency of model training by pruning a large dataset and retaining only a small subset of representative samples. Most CS methods work by first using a score to estimate the difficulty/importance of every training sample and then use a sampling strategy that forms the coreset. Many state-of-the-art (SOTA) CS methods use the downstream model's training dynamics — changes in the model's behavior on a sample over epochs during training, to generate an importance score for each sample. While this enables a good estimation of the data importance, it requires

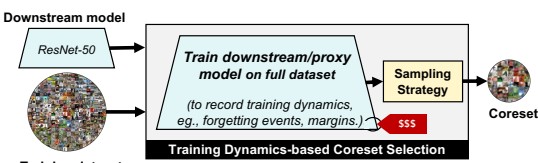

Figure 1: SOTA CS methods assess a sample's difficulty by training a downstream model on the **full** dataset at least once, making them costly. We propose an efficiently computable and interpretable score that **eliminates** the need for this model/its training dynamics.

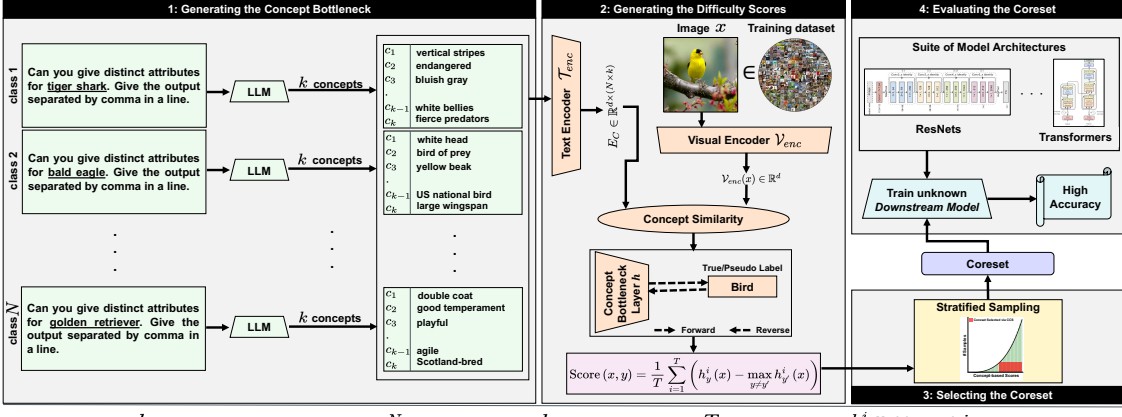

Figure 2: **Overview of our approach:** We start by prompting an LLM to generate concept annotation for $N$ class label names in the dataset and select $k$ most discriminative attributes (per class) to form the concept bottleneck, which are passed through a text encoder ($\mathcal{T}_{enc}$) to obtain the bottleneck embedding matrix ($E_C$). The visual information of a training sample $x$ extracted via the visual encoder ($\mathcal{V}_{enc}$) is then aligned to $E_C$ using a linear concept bottleneck layer ($h$) trained for $T$ epochs. Our difficulty score for a sample $x$ is then computed as the average margin (i.e., the difference between the softmax scores of the correct and other classes) over $T$ epochs. Finally, the coreset is selected via stratified sampling and is used to train downstream models.

training the downstream model on the entire dataset at least once, which can be costly when training a large model on a large dataset (see Fig. 1). While Coleman et al. (2019) showed that a coreset selected using training dynamics of a mid-sized proxy model (eg, ResNet-18) is effective for a larger downstream model (eg, ResNet-50), training even such a model may not be feasible for large datasets. Moreover, since these scores are dependent on training dynamics of a particular downstream model, they are hard to interpret as they do not inform us about the sample's importance for another downstream model (without training it first). Thus, in our work we tackle the following question: *"How to efficiently estimate the importance of training samples for CS in a data centric way that is independent of the downstream model and avoids training that model on the full dataset?*

To address this, we use Concept Bottleneck Models (CBMs) (Koh et al., 2020; Yuksekgonul et al., 2022), which work by mapping a model's input onto a set of human-understandable concepts, referred to as the "bottleneck" and use them to make a prediction. However, CBMs require concept annotation for every sample in the dataset, which can be costly to obtain. Recently (Yang et al., 2023b; Yan et al., 2023) showed that off-the-shelf Large Language Models (LLMs) and Vision Language Models (VLMs) can be prompted to obtain concept annotations for training samples without requiring any task-specific fine-tuning (see Fig. 2 (block 1)). Once the bottleneck is formed, we use a VLM (e.g., CLIP (Radford et al., 2021)) to measure the alignment between the visual features and the concept bottleneck (denoted as concept similarity in Fig. 2 (block 2)). A linear concept bottleneck layer is then trained to align the visual and concept features while classifying samples. We then use the average margin of a sample while training the bottleneck layer as our concept-based importance score. Finally, we form the coreset using stratified sampling (Zheng et al., 2022) (block 3) based on our score, which is used for training downstream models (Fig. 2 (block 4)).

We empirically evaluate the effectiveness of our score on three benchmark datasets: CIFAR-10/100 (Krizhevsky et al., 2009) and Imagenet-1K (Deng et al., 2009), as well as on *biomedical* (Acevedo et al., 2020) and *affective computing* (Mollahosseini et al., 2017) tasks. Our results show that downstream models trained on our coresets consistently achieve better accuracy than randomly sampled subsets, especially at high data pruning rates, and achieve performance close to SOTA CS methods. We also show that our approach is effective for the label-free CS problem where the dataset is unlabeled and leads to models with superior performance compared to SOTA label-free CS methods, especially on Imagenet. Since our CS method is independent of the downstream model, we show that our coreset leads to high performance regardless of the architecture of the downstream model. Moreover, our method speeds up the computation of the coreset by $\approx 8$ *times*

compared to approaches based on the training dynamics of the downstream model. Lastly, we show that our concept-based difficulty score provides an intuitive explanation of why examples are easy/hard, independent of the downstream model. Our main contributions are summarized below:

- We propose a concept-based score that efficiently computes a training sample's importance for CS without training a downstream model on the full dataset, even once.

- Our coresets improve accuracy by $\approx 5\%$ over random subsets at high pruning rates, are competitive to coresets found by SOTA methods, transfer to various architectures, and can be computed for unlabeled training data.

- We show that using CBMs with LLM-generated concepts makes our score interpretable, enabling a *data-centric* solution for identifying coresets.

## 2  Related Work

**Coreset selection (CS).** CS improves the efficiency of model training by selecting a subset of influential samples. Various approaches have been proposed to generate such a subset (Guo et al., 2022). A popular approach uses influence functions (Koh & Liang, 2017; Chatterjee & Hadi, 1986; Liu et al., 2021; Schioppa et al., 2022) which measures the influence of a sample by considering the effect of removing it from the model's training. While effective, these approaches are computationally costly due to their dependence on higher-order derivatives. Approaches that use the dataset's geometric properties such as (Sener & Savarese, 2017; Sorscher et al., 2022; Feldman et al., 2020; Feldman & Langberg, 2011; Huang et al., 2019) are another popular choice for CS . However, the high computational complexity due to their dependence on pairwise distances between the samples prohibits their use on large datasets. Another set of approaches select a subset by either matching the gradients to those computed on the entire dataset (Mirzasoleiman et al., 2020; Killamsetty et al., 2021) or use training dynamics of a model (Toneva et al., 2018; Pleiss et al., 2020; Lewis & Catlett, 1994; Culotta & McCallum, 2005; Paul et al., 2021) to compute the importance of a sample. However, such approaches require repeated training of the downstream model to produce accurate importance scores. In comparison, our approach avoids using any knowledge of the downstream model for computing the difficulty scores.

**Adaptive subset selection.** These works focus of improving the training convergence and efficiency of model training by selecting a new subset of data from the whole dataset, every epoch, while training a downstream model. Thus, unlike our work, these works do not prune the dataset but rather keep selecting small, potentially non-overlapping, subsets every few epochs. Both (Killamsetty et al., 2023; Tukan et al., 2023), propose ways to select subsets every epoch or every few epochs in a way that does not use a downstream model unlike some other works such as GradMatch (Killamsetty et al., 2021), which select subsets based on the downstream model. While our work does not target adaptive subset selection, we evaluated how well our approach performs without any change on this problem in Sec. 5.4. Our results show that, even on for this application, our concept-based score is an effective method.

**Active learning.** assumes an interactive setting with iterative labeling and downstream model dependent updates to the importance of the samples, while our method focuses on one time coreset selection for fixed datasets without assuming the knowledge of the training dynamics of the downstream model. Specifically, active learning (Settles, 2012; Lewis & Catlett, 1994) relies on the performance of the downstream model at the current epoch to make a decision for which samples from the training data should be used next. While in our method, we disentangle the selection of the coreset from the downstream model, and show that the same coreset leads to high performing models for various choices of downstream model architectures, making it efficient (no repeated data selection) and also generalizable (agnostic to downstream model).

**Concept-based interpretability approaches.** Concepts are defined as high-level semantics that refers to the abstract and human-interpretable meanings of the visual data, such as objects, actions, and scenes, as opposed to low-level features like edges or textures (Wu et al., 2016). Concepts have been used in interpretable computer vision to bridge the gap between human understanding and machine perception in various tasks such as image classification. Such interpretability methods can be broadly classified as *post-hoc methods* (do

not impose any model constraints) or *by-design* methods (see App. A). Concept Bottleneck Models (CBMs) extend interpretable-by-design approaches by using human-understandable attributes as an intermediate layer for predictions, as used in few-shot learning (Lampert et al., 2013) and attribute learning (Xu et al., 2020; Russakovsky & Fei-Fei, 2012). While interpretable, CBMs reliance on costly annotations and lower accuracy compared to end-to-end models limit their usage. Post-hoc Concept Bottleneck Models (PCBMs) address these issues by incorporating static knowledge bases (e.g., ConceptNet Speer et al. (2017)) and residual connections to boost accuracy (Yuksekgonul et al., 2022). Recently Yang et al. (2023b); Yan et al. (2023) incorporated LLMs to identify the concept bottleneck making classification more explainable. We build on this and use CBMs for CS.

## 3 Preliminaries

### 3.1 Coreset selection (CS) problem formulation

Consider a classification task and data distribution $P$. Let $\mathcal{D} = \{(x_i, y_i)\}_{i=1}^n$ denote the dataset of $n$ training examples sampled i.i.d. from the distribution $P$ where $x_i$ denotes the data and $y_i \in \mathcal{Y}$ denotes the label from a set of $N$ classes. CS (Coleman et al., 2019; Zheng et al., 2022) aims to find a subset $\mathcal{S}$ of $\mathcal{D}$ consisting of $m \leq n$ samples such that the models trained on $\mathcal{S}$ achieve performance comparable to models trained on $\mathcal{D}$. Let $\theta_{\mathcal{D}}$ and $\theta_{\mathcal{S}}$ denote the "*downstream model*" trained on $\mathcal{D}$ and $\mathcal{S}$ (coreset), respectively and $\ell$ be the loss function then the CS problem is as follows

$$\min_{\mathcal{S}:\mathcal{S}\subset\mathcal{D},|\mathcal{S}|=m} \mathbb{E}_{(x,y)\sim P}[\ell(x,y|\theta_{\mathcal{S}})] - \mathbb{E}_{(x,y)\sim P}[\ell(x,y|\theta_{\mathcal{D}})]. \tag{1}$$

To find this subset $\mathcal{S}$, previous works have proposed scores that gauge a sample's difficulty for a model. Approaches such as max entropy uncertainty sampling (Lewis & Catlett, 1994; Settles, 2012), and least confidence (Culotta & McCallum, 2005) estimate difficulty using the uncertainty of the model's predictions on a sample. Another set of approaches such as $k$-center greedy (Sener & Savarese, 2017) uses geometric information of the data to filter out redundant samples. Yet, another set of approaches uses training dynamics of the downstream model to estimate the difficulty score. Scores such as the forgetting events (Toneva et al., 2018) computed as the number of times a sample gets misclassified after being correctly classified earlier during model training, and the area under the margin (AUM) (Pleiss et al., 2020) which identifies mislabeled/difficult samples, fall in this category.

### 3.2 Concept bottleneck models (CBMs)

Recent advances in language model-guided CBMs utilize an off-the-shelf LLM to obtain concept bottlenecks which are then used to predict the labels. These works rely on pre-trained multi-modal models (such as CLIP (Radford et al., 2021)) which consists of a visual encoder $\mathcal{V}_{enc}$ and a text encoder $\mathcal{T}_{enc}$ that can map images and text to a $d$-dimensional representation space. Let $C = \{c_1, c_2, \cdots, c_{N_C}\}$ be the set of $N_C$ concepts (bottleneck) generated via an LLM, we can then construct a bottleneck embedding matrix $E_C \in \mathcal{R}^{N_C \times d}$ such that each row of the matrix is a mapping of the concept $c \in C$ after passing it through textual encoder $\mathcal{T}_{enc}$. Based on this, a CBM (Yang et al., 2023b) produces a prediction $h(x) = f(g(\mathcal{V}_{enc}(x); E_C))$ for a sample $x$ where $g : \mathbb{R}^d \to \mathbb{R}^{N_C}$ computes the similarity of the visual features to each concept in the bottleneck and $f : \mathbb{R}^{N_C} \to \Delta$ outputs the probability of each class in the label set $\mathcal{Y}$, where $\Delta$ is a $N$ simplex. We discuss details of $f$ and $g$ in Sec. 4.

## 4 Methodology

### 4.1 Generating the concept bottleneck via LLMs

Since obtaining data with concept annotation is costly, we use LLMs to generate concept annotation for the samples. However, generating attributes (word-level concepts) for all the samples in the dataset via LLMs is still costly, hence we generate word-level concepts *only* for class label names. This approach was recently

---

**Algorithm 1** Concept-Based Coreset Selection Pipeline

---

**Require:** Dataset $\mathcal{D} = \{(x_i, y_i)\}_{i=1}^n$, class names $\mathcal{Y}$, VLM, visual encoder $V_{\text{enc}}$, text encoder $T_{\text{enc}}$, concepts-per-class $k$, AUM epochs $T$, CCS pruning ratio $\alpha$, CCS cutoff rate $\beta$, CCS bins $b$.

**Ensure:** Coreset $\mathcal{S}$

1: {Build the concept bottleneck}
2: $(C, E_C, \texttt{bottleneck\_concepts}) \leftarrow \text{GenerateConceptBottleneck}(\mathcal{Y}, k, \text{VLM}, T_{\text{enc}})$ (Alg. 2)
3: {Compute difficulty scores via Eq. 4}
4: $(\{\text{AUM}_i\}_{i=1}^n, W) \leftarrow \text{ConceptBasedAUMScoring}(\mathcal{D}, V_{\text{enc}}, E_C, T)$ (Alg. 3)
5: {Package dataset with sample-wise scores and select coreset using CCS}
6: $\mathbb{D} \leftarrow \{(x_i, y_i, \text{AUM}_i)\}_{i=1}^n$
7: $\mathcal{S} \leftarrow \text{CCS}(\mathbb{D}, \alpha, \beta, b)$ (Alg. 4)
8: Return $\mathcal{S}$

---

shown to be effective at generating the concept bottleneck for interpretable image classification (Yan et al., 2023; Yang et al., 2023b).

In Figure 2 (block 1), we present the prompts provided to the LLMs to extract the concepts for various class label names. The responses of the LLM are then processed to remove formatting errors to obtain the concept sets. Once the per-class concepts are extracted, we select $k$ discriminative concepts per class (concepts that are unique to a class) to form the concept bottleneck. Our final list of $k$ concepts for a class contain its class name and $k - 1$ attributes generated by the LLM. Details of our prompt design, robustness check of different prompts, and examples of the generated attributes are mentioned in App. B.

### 4.2 Concept-based score for CS

We now describe how to use the concept bottleneck to produce a difficulty score for the samples in the dataset. We start by discussing how we learn the functions $f$ and $g$ described in Sec. 3.2 (see Fig. 2 (block 2)). We use the dot product between the visual embeddings of an image $x$ i.e. $\mathcal{V}_{enc}(x)$ and the bottleneck embedding matrix $E_C$ to measure the alignment between the visual and textual features (Yang et al., 2023b; Yan et al., 2023). Concretely, we compute the concept similarity score for a sample $x$ as

$$g(x; E_C) := \mathcal{V}_{enc}(x) \cdot E_C^\mathsf{T}. \tag{2}$$

To map the concept similarity score to a prediction in the label space $\mathcal{Y}$, we propose to use a linear (concept bottleneck layer) predictor denoted by $f$. Concretely, the function $f$ with parameters $W \in \mathbb{R}^{N \times N_C}$ is given by $f(x; W) := g(x; E_C) \cdot W^\mathsf{T}$. We learn the parameters $W$ using

$$W^* = \arg\min_W \frac{1}{n} \sum_{i=1}^n \ell(f(x_i; W), y_i), \tag{3}$$

where $\ell(f(x; W), y) = -\log(f(x; W)_y)$ is the cross-entropy loss. The output of the concept bottleneck layer is defined as, $h(x) := f(g(x; E_C); W^*)$. In practice, we learn $W$ using mini-batch gradient descent by running the optimization for $T$ epochs. We then compute the difficulty of each training sample using area under the margin (AUM) (Pleiss et al., 2020) while solving Eq. 3, quantifying the data difficulty as the margin of a training sample averaged over $T$ training epochs. Concretely, margin of a sample $(x, y)$ at an epoch $t$ is $M^t(x, y) = h_y^t(x) - \max_{y' \neq y} h_{y'}^t(x)$, where $h_{y'}^t(x)$ is the prediction likelihood of the bottleneck layer at epoch $t$ for a class $y'$. Thus, AUM (concept-based score) is

$$\text{AUM}(x, y) = \frac{1}{T} \sum_{t=1}^T M^t(x, y). \tag{4}$$

Recent works (Pleiss et al., 2020; Zheng et al., 2022; 2024) have demonstrated the effectiveness of AUM for computing a sample's difficulty for CS. However, (Zheng et al., 2022; 2024) compute AUM for a specific downstream model by training it on the entire dataset first, which is computationally costly. On the other

hand, we integrate AUM with the training of a linear layer $h$, training which is significantly cheaper than training the downstream model (training $h$ for 100 epochs takes only 7 minutes on Imagenet compared to 8 hours for training a ResNet-34 for 100 epochs). Moreover, since our score is independent of the downstream model, our coresets can be used for any downstream model without change, unlike training dynamics-based approaches that require computing their coresets again for different/new downstream models/architectures.

**Sampling training examples to form a coreset.** After obtaining data difficulty scores, a crucial step is choosing the samples to form the coreset. While many previous works (Toneva et al., 2018; Coleman et al., 2019) have reported encouraging results by keeping only the most challenging samples (for our concept-based score this means samples with the smallest margin), recent works (Zheng et al., 2022; Sorscher et al., 2022) have shown that this could lead to a catastrophic drop in accuracies after training the downstream model on the coreset, especially when the size of the coreset is small. This is mainly due to poor sample coverage and potentially mislabeled data in the datasets. To remedy this, we use Coverage-centric Coreset Selection (CCS) proposed by (Zheng et al., 2022) (see Alg. 4 in App. C.4) which filters out (potentially) mislabeled samples and uses a stratified sampling approach to form the coreset. This technique has been shown to consistently achieve superior results to the random baselines for various coreset sizes. We summarize the entire pipeline in Algorithm 1.

### 4.3 Concept-based score for label-free CS

Recently, there has been an interest (Zheng et al., 2024; Maharana et al., 2023; Griffin et al., 2024) in identifying the representative samples from an unlabeled dataset so as to 1) reduce the samples that need to be labeled/annotated by humans and 2) improve the efficiency of model training by only training the model on a subset of data. Our concept-based score can also be effectively utilized for this task with a simple modification. Similar to previous works (Maharana et al., 2023; Zheng et al., 2024; Sorscher et al., 2022), we assume that we know the number of classes in the datasets. Additionally, we assume that we also know the names of the classes in the datasets. Previous works have demonstrated that VLMs such as CLIP (Radford et al., 2021) achieve excellent zero-shot performance without requiring fine-tuning on specific datasets. We leverage this capability of CLIP models to obtain pseudo-labels for our unlabeled dataset and use them to obtain our difficulty score as follows

$$\text{AUM}(x, y_{\texttt{pseudo}}) = \frac{1}{T} \sum_{t=1}^{T} M^t(x, y_{\texttt{pseudo}}),$$ (5)

where for an image $x$ in the dataset $y_{\texttt{pseudo}} = \arg\max_{j \in \mathcal{Y}} \mathcal{V}_{enc}(x) \cdot \mathcal{W}_{\texttt{zeroshot}}^{\mathsf{T}}$ where $\mathcal{W}_{\texttt{zeroshot}} \in \mathbb{R}^{N \times d}$ is a matrix with columns defined as $\mathcal{T}_{enc}(s_j)$ and $s_j = $ "a photo of a $\{j^{th}$ class name$\}$" for each class $j \in \mathcal{Y}$ (Radford et al., 2021; Wortsman et al., 2022). We use these scores along with CCS (Zheng et al., 2022) to produce the coreset. Similar to (Zheng et al., 2024; Maharana et al., 2023), the coreset is then assumed to be annotated by humans and used for training.

## 5 Experiments

**Datasets, models, and training:** We focus on CS for classification tasks on three benchmark datasets namely, CIFAR-10, CIFAR-100 (Krizhevsky et al., 2009), and Imagenet-1K (Deng et al., 2009) datasets consisting of 50000, 50000, and 1.28 million samples spread across 10, 100, and 1000 classes, respectively. For CIFAR-10/CIFAR-100, we train a ResNet(RN)-18 as a downstream model and for Imagenet we train ResNet-18/34/50 (He et al., 2016), MobileNet (Sandler et al., 2018), DenseNet (Huang et al., 2017), Wide ResNet (Zagoruyko & Komodakis, 2016), and ViT (Dosovitskiy et al., 2020) as downstream models on the coresets. We use FFCV (Leclerc et al., 2023) to accelerate training on Imagenet. We run CS for three trials with different random seeds and report the average of these runs in our tables for various pruning rates where a pruning rate of 90% refers to removing 90% of the samples.

For generating concepts we use a recently proposed open source model LLaVA (Liu et al., 2023b;a). For computing the concept similarity scores between the visual and concept features we used the CLIP (Radford et al., 2021) model (with the ViT B-32 as backbone) following the previous works (Yun et al., 2022; Yang

Table 1: Comparison of the model's (RN-18 for CIFAR10/100 and RN-34 for Imagenet) test accuracy after training on coresets found by various approaches shows that our coresets lead to significantly better performance than Random and achieve competitive results compared to the methods using the downstream model's training dynamics, even for high pruning rates. Results for Forgetting and AUM are taken from (Zheng et al., 2022). Best results in each category based on the need for the knowledge or training dynamics of the downstream model are highlighted. Rows corresponding to methods that require training dynamics are  shaded .

| | Datasets and Pruning Rates | | | | | | | | | | | |
|---|---|---|---|---|---|---|---|---|---|---|---|---|
| **Method** | *CIFAR-10* | | | | *CIFAR-100* | | | | *Imagenet* | | | |
| | 30% | 50% | 70% | 90% | 30% | 50% | 70% | 90% | 30% | 50% | 70% | 90% |
| Entropy (Coleman et al., 2019) | 94.44 | 92.11 | 85.67 | 66.52 | 72.26 | 63.26 | 50.49 | 28.96 | 72.34 | 70.76 | 64.04 | 39.04 |
| Forgetting (Zheng et al., 2022) | **95.40** | **95.04** | 92.97 | 85.70 | **77.14** | **74.45** | **68.92** | **55.59** | **72.60** | **70.89** | 66.51 | 52.28 |
| AUM (Zheng et al., 2022) | 95.27 | 94.93 | **93.00** | **86.08** | 76.84 | 73.77 | 68.85 | 55.03 | 72.29 | 70.52 | **67.78** | **57.36** |
| Random | 94.33 | 93.40 | 90.94 | 79.08 | 74.59 | 71.07 | 65.30 | 44.76 | 72.18 | 70.34 | 66.67 | 52.34 |
| Random$_{\text{FFCV}}$ | - | - | - | - | - | - | - | - | 73.37$_{\pm 0.08}$ | 71.71$_{\pm 0.10}$ | 67.85$_{\pm 0.04}$ | 51.29$_{\pm 0.20}$ |
| **Ours** | **94.77**$_{\pm 0.09}$ | **93.44**$_{\pm 0.61}$ | **91.80**$_{\pm 0.21}$ | **84.63**$_{\pm 0.24}$ | **75.98**$_{\pm 0.26}$ | **72.22**$_{\pm 0.22}$ | **66.53**$_{\pm 0.42}$ | **51.85**$_{\pm 0.29}$ | **73.39**$_{\pm 0.12}$ | **72.34**$_{\pm 0.13}$ | **69.44**$_{\pm 0.17}$ | **55.92**$_{\pm 0.02}$ |

Table 2: Performance of concept based coreset selection on **emotion recognition (AffectNet)** and **biomedical image recognition (BloodMNIST)** tasks. Coresets selected by our approach improve F1 score and accuracy at high pruning rates and perform similar or better than methods that require access to the downstream model or its training dynamics. Rows corresponding to methods that require training dynamics are  shaded .

| | Datasets and Pruning Rates | | | | | | | |
|---|---|---|---|---|---|---|---|---|
| **Method** | AffectNet (F1 Score) | | | | BloodMNIST (Accuracy) | | | |
| | 30% | 50% | 70% | 90% | 30% | 50% | 70% | 90% |
| Forgetting (Zheng et al., 2022) | **0.602**$_{\pm 0.006}$ | 0.570$_{\pm 0.012}$ | **0.540**$_{\pm 0.013}$ | 0.416$_{\pm 0.005}$ | 95.51$_{\pm 0.067}$ | 94.88$_{\pm 0.215}$ | 94.08$_{\pm 0.691}$ | **84.46**$_{\pm 1.009}$ |
| AUM (Zheng et al., 2022) | 0.595$_{\pm 0.005}$ | **0.596**$_{\pm 0.013}$ | 0.536$_{\pm 0.003}$ | **0.439**$_{\pm 0.012}$ | **95.75**$_{\pm 0.289}$ | **95.54**$_{\pm 0.271}$ | **94.72**$_{\pm 0.425}$ | 81.22$_{\pm 2.038}$ |
| Random | **0.607**$_{\pm 0.006}$ | 0.573$_{\pm 0.025}$ | 0.517$_{\pm 0.006}$ | 0.347$_{\pm 0.055}$ | **94.99**$_{\pm 0.267}$ | **94.79**$_{\pm 0.176}$ | 91.71$_{\pm 0.286}$ | 86.50$_{\pm 0.655}$ |
| Ours | 0.603$_{\pm 0.006}$ | **0.577**$_{\pm 0.021}$ | **0.537**$_{\pm 0.015}$ | **0.450**$_{\pm 0.035}$ | 94.64$_{\pm 0.599}$ | 94.06$_{\pm 0.096}$ | **92.26**$_{\pm 0.190}$ | **87.44**$_{\pm 1.455}$ |

et al., 2023b; Yan et al., 2023). For computing the pseudo-labels for label-free CS (in Sec. 4.3) we used a ViT L-14 CLIP model trained on the DataComp-1B dataset (Ilharco et al., 2021). We present ablations of various choices for LLM/VLMs for concept extraction and similarity computation in Sec. 5.4. Further experimental details are mentioned in App. C.

**CS baselines and methods:** We compare our method against various baselines and SOTA CS methods. 1) **Random:** Uniformly select samples from the datasets to form the coreset. Random$_{\text{FFCV}}$ denotes the performance of the models trained on random subsets of Imagenet using FFCV (Leclerc et al., 2023). 2) **Entropy (Coleman et al., 2019):** Selects samples based on entropy computed as the uncertainty of a model's prediction on a sample. 3) **Forgetting (Toneva et al., 2018):** Selects samples based on the forgetting score computed as the number of times a sample is misclassified after being correctly classified earlier during training of the downstream model. A higher forgetting score indicates a more challenging sample. 4) **AUM (Pleiss et al., 2020):** Selects samples based on their average margin during training of the downstream model i.e., the difference between the target class and the next highest class across the training epochs. Lower AUM indicates a more challenging sample. For forgetting, AUM, and our method, we use CCS (Zheng et al., 2022) to form the coreset whereas for entropy we select samples with the highest entropy as done in (Coleman et al., 2019; Zheng et al., 2022).

For label-free CS, we use 1) **Prototypicality (Sorscher et al., 2022):** which first performs k-means clustering in the embedding space of SwAV (Caron et al., 2020) model and ranks samples based on their Euclidean distance to the cluster centers. Samples further away from the cluster center are used to form the coreset. 2) **ELFS (Zheng et al., 2024):** estimates the pseudo-labels of the unlabeled samples using a deep clustering approach (using the embedding space of SwAV (Caron et al., 2020) and DINO (Caron et al., 2021)) and forms the coreset using training dynamics of the downstream model trained on the pseudo-labeled data. Crucially, SOTA methods such as forgetting, AUM, and ELFS train the downstream model on the entire dataset first for CS, unlike our method which is independent of the downstream model. While Random

Table 3: Comparison of the model's test accuracy after training on coresets, found in a **label free** manner, shows that our coresets lead to better performance than Random and Prototypicality. Our coresets also achieve performance competitive to or better than the coresets found by the state of the art method ELFS(Zheng et al., 2024), which relies on the training dynamics of the downstream model. Rows corresponding to methods that require training dynamics are shaded .

| Method | Datasets and Pruning Rates | | | | | | | | | | | |
| --- | --- | --- | --- | --- | --- | --- | --- | --- | --- | --- | --- | --- |
| | *CIFAR-10* | | | | *CIFAR-100* | | | | *Imagenet* | | | |
| | 30% | 50% | 70% | 90% | 30% | 50% | 70% | 90% | 30% | 50% | 70% | 90% |
| ELFS (SwAV) (Zheng et al., 2024) | 95.00 | 94.30 | 91.80 | 82.50 | 76.10 | 72.10 | 65.50 | 49.80 | 73.20 | 71.40 | 66.80 | 53.40 |
| ELFS (DINO) (Zheng et al., 2024) | **95.50** | **95.20** | **93.20** | **87.30** | **76.80** | **73.60** | **68.40** | **54.90** | **73.50** | **71.80** | **67.20** | **54.90** |
| Prototypicality (Sorscher et al., 2022) | 94.70 | 92.90 | 90.10 | 70.90 | 74.50 | 69.80 | 61.10 | 32.10 | 70.90 | 60.80 | 54.60 | 30.60 |
| Random | 94.33 | 93.40 | 90.94 | 79.08 | 74.59 | 71.07 | 65.30 | 44.76 | 72.18 | 70.34 | 66.67 | 52.34 |
| Random$_{FFCV}$ | - | - | - | - | - | - | - | - | 73.37 ±0.08 | 71.71 ±0.10 | 67.85 ±0.04 | 51.29 ±0.20 |
| **Ours-LF** | **94.81** ±0.14 | **93.93** ±0.13 | **91.75** ±0.34 | **84.02** ±0.44 | **74.67** ±0.23 | **72.07** ±0.58 | **65.50** ±0.17 | **49.91** ±0.96 | **73.61** ±0.08 | **71.99** ±0.05 | **68.42** ±0.21 | **53.21** ±0.06 |

and Prototypicality also don't require the downstream model for CS, our results show that our approach surpasses them.

## 5.1 Evaluating performance of our score for CS

**Effectiveness on standard CS.** Table 1 shows the accuracy of models trained on coresets found by various approaches on the **standard** CS problem (where the dataset is labeled). Our results show that our coresets lead to significantly better performance, even at higher pruning rates, compared to random subsets. Moreover, our method, which does not have the knowledge of the downstream model or its training dynamics, provides competitive performance to coresets found by the SOTA approaches based on forgetting and AUM, and even outperforms them on Imagenet for smaller pruning rates.

**Effectiveness on biomedical and affective computing tasks.** Here, we present an evaluation of using our approach for CS for emotion recognition (using a subset of AffectNet (Mollahosseini et al., 2017)) and biomedical entity recognition (using BloodMNIST (Acevedo et al., 2020)) task. Superior performance of our approach in Table 2 compared to Random highlights the effectiveness of our approach for CS on diverse tasks. Our results also show that our method which does not requires access to the downstream model or it's training dynamics achieves results comparable to methods such as forgetting and AUM on these tasks. Further details of this experiment are presented in App. C.2.

**Performance on label-free CS.** Table 3 shows the accuracy of models trained on coresets for **label-free** CS, where the training set is unlabeled (we report the numbers presented by (Zheng et al., 2024) for previous methods). The results show that the random subsets are a competitive baseline for this problem outperforming Prototypicality (Sorscher et al., 2022). Our results also show that our coresets outperform the random subsets for all pruning rates with significant improvements at higher pruning rates. Compared to ELFS (Zheng et al., 2024), our method provides competitive performance and even surpasses it for lower pruning rates on Imagenet, without using any information about the downstream model's architecture or its training dynamics highlighting its effectiveness.

**Transferability of our coresets to various model architectures.** Next, we evaluate the **transferability** of the coresets found by our approach to downstream models with different architectures including those based on convolutional neural networks, ResNets and ViTs. The results in Table 4 and Table 14 (in App. C.3) show that our coresets achieve superior performance over random for all the architectures. This results highlights that our method does not need to recompute the coresets for a new downstream model architecture, unlike training dynamics based approaches which identify the best performing coreset by using information of the downstream model.

Thus, our concept-based score in conjunction with stratified sampling (Zheng et al., 2022) is an effective approach for standard and label-free CS at various pruning rates and leads to high performing and transferable coresets for a diverse set of tasks.

Table 4: Superior performance of downstream models with various architectures trained on our coresets for Imagenet compared to Random for standard (Ours) and label-free (Ours-LF) CS highlights the transferability of our coresets to various models.

| Model Architecture | Method | Pruning Rates | | | |
|---|---|---|---|---|---|
| | | 30% | 50% | 70% | 90% |
| **RN-18** | **Random** | $71.15_{\pm 0.23}$ | $68.48_{\pm 0.10}$ | $63.15_{\pm 0.19}$ | $44.96_{\pm 0.50}$ |
| | **Ours-LF** | $71.21_{\pm 0.09}$ | $68.77_{\pm 0.13}$ | $63.76_{\pm 0.09}$ | $47.50_{\pm 0.10}$ |
| | **Ours** | $70.94_{\pm 0.19}$ | $69.30_{\pm 0.84}$ | $65.16_{\pm 0.04}$ | $49.57_{\pm 0.16}$ |
| **RN-50** | **Random** | $76.06_{\pm 0.11}$ | $74.44_{\pm 0.04}$ | $70.50_{\pm 0.02}$ | $53.56_{\pm 0.13}$ |
| | **Ours-LF** | $76.54_{\pm 0.08}$ | $74.84_{\pm 0.03}$ | $71.10_{\pm 0.09}$ | $55.22_{\pm 0.92}$ |
| | **Ours** | $76.29_{\pm 0.10}$ | $75.12_{\pm 0.03}$ | $72.05_{\pm 0.09}$ | $58.26_{\pm 0.46}$ |
| **DenseNet** | **Random** | $76.55_{\pm 0.11}$ | $75.31_{\pm 0.15}$ | $71.85_{\pm 0.20}$ | $56.27_{\pm 0.21}$ |
| | **Ours-LF** | $76.94_{\pm 0.23}$ | $75.52_{\pm 0.11}$ | $72.28_{\pm 0.14}$ | $58.16_{\pm 0.04}$ |
| | **Ours** | $76.80_{\pm 0.22}$ | $75.94_{\pm 0.03}$ | $73.40_{\pm 0.27}$ | $60.93_{\pm 0.19}$ |
| **Wide Resnet** | **Random** | $77.39_{\pm 0.19}$ | $75.32_{\pm 0.12}$ | $71.51_{\pm 0.10}$ | $54.78_{\pm 0.83}$ |
| | **Ours-LF** | $77.83_{\pm 0.08}$ | $75.77_{\pm 0.09}$ | $71.97_{\pm 0.08}$ | $57.34_{\pm 0.58}$ |
| | **Ours** | $77.48_{\pm 0.05}$ | $76.32_{\pm 0.03}$ | $73.14_{\pm 0.03}$ | $59.75_{\pm 0.70}$ |

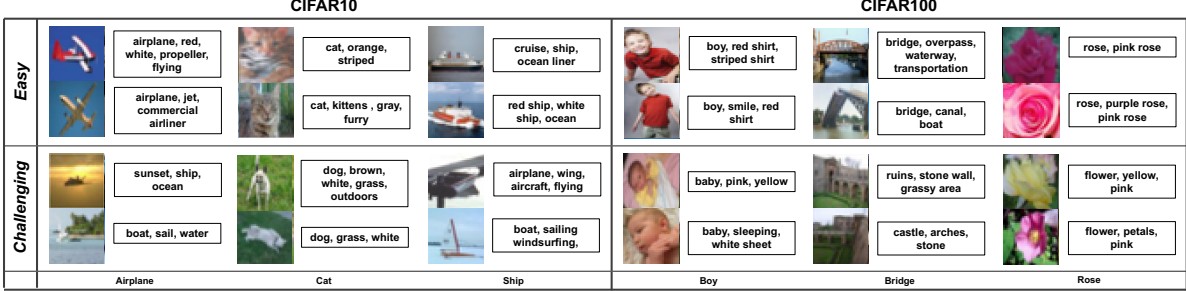

Figure 3: Visualizing samples according to our concept-based score for a subset of classes in CIFAR-10/100 showing that easy (challenging) samples are aligned (unaligned) with their assigned label. Image-level concepts (in boxes) extracted via LLaVA confirm that easy (challenging) examples are aligned (unaligned) with concepts of their labels, explaining the reason for a high (low) concept-based score.

## 5.2 Evaluating efficiency of our score for CS

Here, we compare the efficiency of our approach in finding coresets compared to approaches based on training dynamics. For training dynamics-based approaches, the time required to find the coreset is dominated by the time needed to train the downstream model on the entire dataset first. For example, finding a coreset of Imagenet using a ResNet-34 model takes roughly 8 hours on two A-100 GPUs. In comparison, for our approach, extracting concepts via LLMs (block 1 of Fig. 2) takes 3 seconds per class (totaling to 25 minutes for all classes of Imagenet using 2 GPUs), and computing visual/concept features via CLIP and training a linear concept bottleneck layer (block 2 for Fig. 2) takes roughly 30 minutes for Imagenet offering $\approx 8x$ speedup for CS over approaches relying on training dynamics of the downstream model. Moreover, since our method is independent of the downstream model architecture, we do not need to repeat the CS step for different architectures. This is in contrast with training dynamics-based methods that require training the downstream model for every new architecture to find the corresponding best coreset for it.

## 5.3 Visualizing easy and challenging samples based on our concept-based score

Here we show the advantage of our concept-based score for *assessing the sample's difficulty and how using concepts aids its interpretability*. We start by visualizing the easiest and the most challenging images (per class) for CIFAR-10/100. In Fig. 3, the top row shows the images with the highest concept-based scores (easiest) and the bottom shows the images with the lowest scores (challenging) for a subset of classes in CIFAR-10/100. As observed the easiest images are typical images associated with the label where as the challenging images are confusing (and even potentially mislabeled) as they look like images from a different

Table 5: Effect of concepts via text only (attributes vs. descriptions) to form the concept bottleneck layer[1].

|  | class-wise attributes | class-wise descriptions |
|---|---|---|
| **Ours** | $\mathbf{51.85}_{\pm 0.29}$ | $51.05_{\pm 0.71}$ |
| **Ours-LF** | $49.91_{\pm 0.96}$ | $49.85_{\pm 0.83}$ |

Table 6: Effect of concepts via both visual and textual information to form the concept bottleneck layer[1].

|  | class-wise one-shot image attributes | image-wise attributes |
|---|---|---|
| **Ours** | $51.68_{\pm 0.45}$ | $52.47_{\pm 0.39}$ |
| **Ours-LF** | $50.22_{\pm 0.16}$ | $51.05_{\pm 0.93}$ |

class. For example, some challenging images in the class "boy" from CIFAR-100 are actually images of a baby which is also a class in CIFAR-100. Similarly, some challenging images from the class "cat" in CIFAR-10 look like images of a dog. More examples of such images are presented in Fig. 4 in App. C.1. Since the challenging examples seem to be confusing, it shows that our score can identify examples that are ambiguous or mislabeled. Such samples may be hard for some ML models to learn and could force them to rely on spurious features, potentially hurting generalization. The ability of our score to identify such samples without the downstream model demonstrates its effectiveness for ranking the samples for CS.

Next, we demonstrate *why certain samples get low/high concept-based scores* in our approach by extracting concepts specific to these images using LLaVA (note that these concepts are different from the per-class concepts used in the concept bottleneck). To generate these, we prompt LLaVA to produce concepts using both the sample's image and its class label (see image-level concept extraction in App. B). These image-level concepts are shown in the boxes in Fig. 3. As observed in Fig. 3, image-level concepts provided by LLaVA are related to the class label for easy images whereas they are unrelated for challenging images. For example, attributes provided by LLaVA for the challenging images of "airplane" align more with those of ship (both which are classes in CIFAR-10), and concepts provided for challenging images of "bridges" align more with those of castles (both of which are classes in CIFAR-100). Since our concept-based score in Eq. 4 assigns a small value for these images, we see that it correctly captures when a sample's visual information is not aligned with the sample's associated label and vice-versa. Thus, explaining why certain examples should be included/excluded from the coreset in an interpretable way independent of the downstream model.

### 5.4 Analysis and ablation studies

Here, we present an analysis and ablation studies to evaluate various components of our approach. Specifically, we evaluate the effect of different 1) methods and LLMs used for concept generation, 2) number of concepts per class ($k$) in the bottleneck, 3) CLIP backbones for visual/concept similarity, 4) number of training iterations ($T$) and size of the concept bottleneck model. We also present an analysis of using different sampling strategies for forming the coreset along with evaluation of our method for the adaptive coreset selection problem. Here we use CIFAR-100 with a 90% pruning ratio and train a ResNet-18 on the selected coresets. Additional ablations on hyperparameters of Alg. 4 are presented in App. C.4.

**Comparison of different techniques to generate concepts via LLaVA.** Tables 5 and 6 shows how the performance of models trained on our coresets change when different method are used to generate the concept sets (Fig. 2 block 1). Since our method uses LLaVA (Liu et al., 2023b), which is a VLM, we compare the performance of models trained on the random subsets and coresets obtained using class-wise concepts (only textual information) and concepts extracted using both visual and textual information.

For concepts generated using only textual information, we consider two alternatives, namely **class-wise attributes** (CW-A) and **class-wise descriptions** (CW-D). While CW-A considers concepts formed by a single or a few words, CW-D consists of longer, more descriptive concepts (eg., a descriptive concept for the class butterfly is "*a beautiful insect with colorful wings*"). For CW-D, we use a subset of $k$ concepts provided by Yang et al. (2023b), generated via the GPT-3 model. Our results show that CW-A performs better than CW-D for both the standard/label-free CS problems. Thus, we used CW-A for all our experiments.

Next, for generating concepts using both visual and textual information, we consider two alternatives. The first is a **class-wise one-shot image attribute** approach where we first cluster all images of a class in the embedding space of the CLIP's visual encoder and identify the image whose embedding is the closest to the cluster center (for the label-free setting we use the pseudo-labels of the images during clustering), then

Table 7: Effect of the number of per-class concepts $k$ and the method of selecting $k$ concepts from those generated by LLaVA on model accuracy[1].

| Concept selection | $k$=1 | $k$=5 | $k$=10 |
|---|---|---|---|
| Random concepts | $48.78_{\pm 0.96}$ | $50.15_{\pm 1.64}$ | $50.39_{\pm 0.72}$ |
| Discriminative | $51.42_{\pm 0.18}$ | $\mathbf{51.85_{\pm 0.29}}$ | $51.22_{\pm 0.72}$ |

Table 8: Effect[1] of using different LLMs for forming the concept bottleneck.

| LLM | Accuracy |
|---|---|
| GPT-3 | $51.05_{\pm 0.71}$ |
| Phi-3 | $51.30_{\pm 0.26}$ |
| LLaVA | $\mathbf{51.85_{\pm 0.29}}$ |

we prompt LLaVA to generate attributes using this single image and the class name. Once generated, we use $k$ discriminative concepts to form the bottleneck. The second alternative is the **image-wise** attribute approach, where we use *each* image in the training set and prompt LLaVA to generate per image attributes describing the image. Once generated, we sort the concepts based on their frequency of occurrence in a class and use the most frequently occurring discriminative concepts to form the bottleneck. While the image-level concepts lead to the best coresets, it is slow and costly to prompt LLaVA to generate attributes for all the images in a large dataset such as Imagenet.

For CIFAR-100, this process took about nine hours to complete (in comparison CW-A can be extracted in 5 minutes without parallel computation) which is very costly compared to the small performance gains it provides over other approaches. Lastly, while the class-wise one-shot image attribute approach is better than CW-A, the additional step of clustering can be costly for larger datasets such as Imagenet. Thus, we use CW-A for concept generation using LLaVA.

**Effect of $k$ and the method for selecting $k$ concepts.** In Table 7, we show how the number of concepts extracted per class label, for creating the bottleneck in block 1 of Fig. 2 affects the selection of coresets. Once the list of class-wise attribute-level concepts is generated by LLaVA, we can select $k$ concepts per class either randomly or choose concepts unique to a class (discriminative). Our results show that using even $k = 1$ is sufficient to surpass the performance using a random subset[1]. This performance increases when we keep discriminative concepts in our concept bottleneck, with $k = 5$ achieving the best results. While the size of the concept bottleneck need not be very large, it is helpful to take a sample's visual similarity with a set of concepts rather than a single concept per class. Thus, we used 5 discriminative concepts per class to form the bottleneck.

**Effect of using different LLMs/VLMs for concept extraction.** We evaluated three different models for concept extraction in block 1 of Fig. 2. We used GPT-3 and two open source VLMs namely Phi-3-Mini-4K-Instruct with 3.8 billion parameters, and LLaVA with 7 billion parameters. In Table 8, we find that the performance of our method remains stable regardless of the LLM used, indicating that even smaller LLMs are effective at generating concepts that produce high performing coresets with our score.

**Effect of using different VLMs for measuring visual and concept similarity.** Here we evaluated different CLIP backbones to compute the similarity between the visual and concept features used in Eq. 2 (block 2 of Fig. 2). Our results in Table 9 show that our concept-based method achieves significantly better performance than Random[1] for all the backbones with ViT B-32 backbone performing the best. Thus, we used this backbone for all the experiments.

**Effect of number of epochs $T$.** Here we evaluated the effect of using different number of epochs $T$ used for training the concept bottleneck layer for computing the concept-based score. Our results in Table 10 show that $T \geq 50$ is enough to achieve concept-based scores that lead to selection of high performing coresets. We used $T = 100$ in our experiments since that achieves the best performance.

**Effect of the size of the CBM model** Here, we evaluate the effect of using different number of layers as a part of model $f$ as described in Sec. 4. Specifically, we tested a model with single, two and three layer fully connected neural networks (FCNNs) with ReLU activations between layers to introduce non-linearity. As shown in the Table 11, increasing the model complexity (non-linearity) does not lead to significant improvements in the performance of the selected coresets. Thus, we used $f$ with a single linear layer.

**Alternate sampling strategies for CS.** In this section, we present an evaluation of using two alternative strategies for identifying the coreset (instead of using stratified sampling). The first one selects the most

Table 9: Effect[1] different backbones for visual-concept similarity in Eq. 2.

| CLIP backbone | Accuracy |
|---|---|
| **ViT B-16** | $48.89_{\pm 1.01}$ |
| **ViT L-14** | $49.54_{\pm 1.82}$ |
| **ResNet-50** | $50.56_{\pm 1.67}$ |
| **ViT B-32** | $\mathbf{51.85}_{\pm 0.29}$ |

Table 10: Effect[1] of number of training epochs $T$ for computing the score in Eq. 4.

| $T$ | Accuracy |
|---|---|
| **10** | $46.63_{\pm 0.08}$ |
| **20** | $48.40_{\pm 0.10}$ |
| **50** | $50.50_{\pm 0.31}$ |
| **100** | $\mathbf{51.85}_{\pm 0.29}$ |

Table 11: Ablation of number of layers in $f$ [1].

| Network | CIFAR-100 |
|---|---|
| Linear model | $\mathbf{51.85 \pm 0.29}$ |
| Two layer FCNN | $50.81 \pm 0.86$ |
| Three layer FCNN | $51.01 \pm 1.21$ |

challenging samples and the second one selects only the easiest samples. We present an evaluation of these strategies using a pruning ratio of 90% on CIFAR-100 in Table 12.

Similar to the finding in previous works (Sorscher et al., 2022; Zheng et al., 2022), we find that for high pruning rate, keeping only the most challenging samples makes it harder for the model to generalize due to presence of less data. Similarly keeping just the very easy samples doesn't give the model enough signal due to lack of data diversity. By contrast, stratified sampling which selects from each difficulty strata yields a high-performing coreset. This finding aligns with the results in Table 4 of Zheng et al. (2022), where even SOTA training dynamics-based methods like Forgetting and AUM underperform random sampling when paired with poor sampling strategies (e.g., selecting only the most difficult examples).

Table 12: Comparing sampling strategies for CS[1].

| Need training dynamics? | Method | Performance |
|---|---|---|
| **No** | Random | 44.76 |
| **Yes** | Forgetting (most challenging) | 15.93 |
| | AUM (most challenging) | 8.77 |
| | Forgetting (CCS) | **55.59** |
| | AUM (CCS) | 55.03 |
| **No** | Ours (most challenging) | 12.97 |
| | Ours (most easy) | 33.81 |
| | Ours (CCS, in the paper) | **51.85** |

**Effectiveness on adaptive subset selection.** In this section, we present a comparison of using our concept-based method for the problem of adaptive subset selection (Killamsetty et al., 2021; 2023; Tukan et al., 2023) (See App. A for a description of the problem). To test our approach, we followed the experimental setup of Tukan et al. (2023), of training a ResNet-18 model on CIFAR-10/100 for 300 epochs and changing the subset every 20 epochs. Unlike works in the line of adaptive subset selection which focus on selecting subsets that do not overlap or train models on easy subsets first before moving to difficult ones, we simply used the CCS-based selection approach (Zheng et al., 2022) to identify new subsets every 20 epochs for this task. Due to the randomness in CCS (Alg. 4), samples being selected to form the coreset changes.

Table 13: Results of our approach for the **adaptive coreset selection** problem highlight its effectiveness on this problem. (Results for RBFNN and GradMatchPB are taken from tables 1 and 3 of Tukan et al. (2023).)

| | Datasets and Pruning Rates | | | |
|---|---|---|---|---|
| **Method** | *CIFAR-10* | | *CIFAR-100* | |
| | 70% | 90% | 70% | 90% |
| GradMatchPB (Killamsetty et al., 2021) | 91.89 | 90.01 | 72.57 | 60.39 |
| RBFNN (Tukan et al., 2023) | 94.44 | 91.40 | 73.48 | 64.59 |
| Adaptive Random | $93.64_{\pm 0.21}$ | $90.49_{\pm 0.41}$ | $71.70_{\pm 0.30}$ | $61.24_{\pm 0.37}$ |
| Ours (adaptive) | $92.61_{\pm 0.08}$ | $89.95_{\pm 0.26}$ | $71.74_{\pm 0.33}$ | $63.71_{\pm 0.48}$ |

In Table 13, we compare our method to previous methods and with an adaptive random selection method (where a random subset of data is selected every 20 epochs), which has been suggested by (Killamsetty et al., 2023) as a strong baseline for this line of work. We observe that our method without any modifications achieves comparable performance to existing adaptive subset selection methods showing its effectiveness for this problem as well. Moreover, sampling new subsets via CCS (Alg. 4) is as efficient as sampling a random subset; thus our approach is also an efficient solution for this problem. While we believe that incorporating better subset sampling techniques as suggested by Killamsetty et al. (2023) may boost the performance of our method on this problem, CCS-based sampling is already quite effective.

---

[1]Cf. A random subset of CIFAR-100 achieves an accuracy of $44.76_{\pm 1.58}$ at 90% pruning rate.

# 6 Conclusion

CS finds representative samples from a large dataset, training models on which leads to models with accuracy similar to the models trained on the entire dataset. In this work, we proposed a scoring mechanism based on concept bottlenecks that allows us to compute the difficulty of a sample in terms of interpretable concepts. This method is independent of the downstream model and avoids training the downstream model on the full dataset even once. Our experiments show that training downstream models on coresets selected using our approach leads to better performance than random subsets and achieves accuracy similar to or better than the SOTA approaches based on training dynamics of the downstream model, for both the standard and label-free CS problem. Moreover, our score provides an intuitive explanation of the difficulty of a sample at the dataset level, independent of any downstream model.

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

# Appendix

We present additional related work in Appendix A. Then we describe the details of our methodology for extracting the concepts from LLaVA in Appendix B and present additional experiments and implementation details of our experiments in Appendix C including the algorithm for stratified sampling used in our work in Appendix C.4.

## A  Additional related work

**Concept-based interpretability:** Interpretability methods can be broadly classified as *post-hoc methods* (do not impose any model constraints) or *by-design* methods. Post-hoc methods include Gradient-weighted Class Activation Mapping approaches (Bau et al., 2017; Selvaraju et al., 2017; Mu & Andreas, 2020; Hernandez et al., 2021) that trace network gradients to identify the input areas that guide predictions and Explanation Generation methods (Singh et al., 2023; Nishida et al., 2022; Kim et al., 2018; Hendricks et al., 2016) that require models to produce explanations for visual tasks by conditioning their predictions on captioning models or incorporating visual evidence to ground explanations (Hendricks et al., 2018; Park et al., 2018). Interpretable-by-design methods, such as Prototype methods, optimize a metric space where classifications are based on distances to class prototypes, identifying important input regions but often obscuring their semantic content (Nauta et al., 2021; Chen et al., 2019; Snell et al., 2017; Satorras & Estrach, 2018; Vinyals et al., 2016).

Concept Bottleneck Models (CBMs) are a part of interpretable-by-design approaches that use human-understandable attributes as an intermediate layer for predictions. A recent advancement, Computational Derivation Learning (CompDL), utilizes a CBM architecture by applying a linear layer over CLIP scores between expert-designed concepts and images, improving evaluation of how well CLIP grounds concepts (Yun et al., 2022). Post-hoc Concept Bottleneck Models (PCBMs) were recently proposed to ease the requirement of CBMs to rely on costly concept annotations and improve their accuracy compared to end-to-end models. However, PCBMs are limited by the coverage of knowledge bases, making them unsuitable for large-scale or domain-specific tasks or fine-grained classification, and their residual predictors can undermine interpretability by blending CBMs with end-to-end models.

High-level semantic-driven descriptions are also used to guide data augmentation to build an informative set (Wickramanayake et al., 2021) to make model training efficient with a good enough training set. Prior works use external knowledge bases to obtain these textual semantic concepts to guide vision models (Bujwid & Sullivan, 2021; Kil & Chao, 2021; Roth et al., 2022; Shen et al., 2022). Thus, the use of concepts has been shown to improve interpretability in various domains. However, to the best of our knowledge, we are the first ones to propose a concept-based score for the CS problem and show its competitiveness to SOTA model training dynamics-dependent approaches.

## B  Details for concept set generation

**Prompt Selection:** To extract concepts for our approach, we only use the class labels in the prompt as can be seen in Figure 2. The prompt, "Can you give distinct attributes for ⟨class name⟩. Give the output separated by a comma in the line." instructs the VLM not only to provide distinct keywords but also adds formatting instructions. However, despite the instructions included in the prompt, LLaVA outputs are not always formatted well, often containing duplicate entries, mismatched commas and braces, and sometimes having a detailed explanation before the keywords. To remedy this we run the LLaVA output through a simple post-processing script and use regular expressions to clean the LLaVA outputs.

For our experiments where we perform ablation of various concept-bottleneck generation methods (Tables 5, 6), we also use two more concept generation methods, one is one-shot image-based class concepts and the second is image-level concept generation. For the former, where we select one representative image per class via clustering, we prompt LLaVA as follows, "⟨image⟩ Can you give distinct attributes for such an image of ⟨class name⟩. Give the output separated by a comma in the line." And, to get concepts for every image of a class, we use a similar prompt as follows, "⟨image⟩ Can you give distinct visual attributes for this image of

⟨class name⟩. Give the output separated by a comma in the line." Each LLaVA prompt request on a single A-100 GPU takes approximately 3 seconds.

**Alternative VLMs for Concept Generation:** We leverage LLaVA as our choice of VLM for concept generation, however in Table 8, we also compared against concepts extracted from GPT (Yan et al., 2023) and another smaller open source VLM (Phi-3-Mini-4K-Instruct). We see comparable performance of our CS method with concepts extracted from these models. We also experimented with retrieving concepts via another recent method, SpLiCE (Bhalla et al., 2024) which uses a linear optimization for sparse concept decomposition. However, a major limitation of SpLiCE is that similar to image-wise attributes as used in Table 6 it is a costly approach (SpLiCE can take up to 3 hours for $50,000$ images, significantly slower than generating class-level concepts from LLaVA).

Moreover, to validate the effectiveness of our coreset selection method, we used concepts provided by a recent work (Xiong et al., 2025). Even with externally generated concepts, our method achieves significantly better performance than random selection. Specifically, on CIFAR-100 with a 90% pruning rate, our approach achieved $50.92 \pm 0.70\%$ accuracy compared to $44.96 \pm 1.58\%$ for random subsets.

## C  Additional experiments and implementation details

### C.1  Visualizing easy/challenging samples based on concept-based score

Similar to Fig 3 in Sec. 5.3 of the main paper, we visualize easy and challenging examples in Fig. 4 for CIFAR-10 and subset of classes from CIFAR-100. As observed the easy images (the ones that get high scores in our approach) are more canonical images of the class labels whereas the challenging ones are images that can potentially be assigned another class in the same dataset or are mislabeled in the dataset. The clear distinctions between these images show that our concept-based score aligns well with human intuition on the difficulty of the samples.

### C.2  Concept-based CS for emotion recognition and biomedical image recognition

To validate the effectiveness of our concept-based coreset selection method beyond object recognition tasks, we apply our concept-based CS approach to the task of emotion recognition and biomedical image recognition.

For emotion recognition, we use the Affectnet dataset (Mollahosseini et al., 2017) for our experiments. AffectNet is a large-scale facial expression dataset designed for training and evaluating affective computing models (Wang et al., 2022). It contains facial images collected from the internet using web search queries for emotion-related keywords in multiple languages. Each image is manually annotated for eight discrete emotion categories: `neutral, happiness, sadness, surprise, fear, disgust, anger, contempt`. For our experiments, we utilize an openly available version of this dataset [2], containing roughly 16000 training and 14000 testing samples.

According to our approach we first use LLaVA to extract concepts for the 8 emotion classes, using the following prompt, *"What are the facial features that distinguish emotion class name from other emotion types. Focus on changes in eyes, nose, lips, eyebrows, mouth. Give the output separated by commas in a line.".* We get $5 - 10$ distinctive facial feature concepts for every emotion, for instance for emotion class *happy*, we get the following concepts, *"wide open eyes", "sparking eyes", "smiling lips", "open mouth", "raised eyebrows", "flushed cheeks", "teeth barred".* We finally select $k = 5$ discriminative concepts from this list.

To test coreset performance, we use the EfficientNet model (Tan & Le, 2019) and report F1 scores for our coresets in Table 2. When compared against randomly selected coresets for the various pruning ratios, coresets selected via our concept-based approach achieve better performance at various pruning rates while achieving competitive performance to methods based on training dynamics.

For biomedical image recognition, we use the BloodMNIST (Acevedo et al., 2020) dataset from MedM-NIST (Yang et al., 2021; 2023a) which comprises of images of normal blood cells, captured from individuals without infection, hematologic or oncologic disease and free of any pharmacologic treatment at

---

[2]https://www.kaggle.com/datasets/noamsegal/affectnet-training-data

the moment of blood collection. It consists of a total of $17,092$ images and is organized into 8 classes (`basophil,eosinophil,erythroblast,`
`immature granulocytes(myelocytes, metamyelocytes and promyelocytes),`
`lymphocyte,monocyte,neutrophil,platelet`).

For this dataset, we first extract concepts for the 8 blood cell types via GPT using the following prompt, *"What are the features that can distinguish blood cell class name from rest of the blood cell types on their size, shape, nucleus appearance, and the presence of granules in their cytoplasm"*. We obtain 10 concepts for every blood cell type, for instance, for *platelets*, we get the following concepts, *"Smallest blood component", "No nucleus", "Granules present", "Irregular shape", "Cytoplasmic fragments", "Variable granule distribution", "Oval to round shape", "Small dense granules", "Lacks chromatin", "Compact cytoplasmic body"*.

To test the coreset performance, we use a ResNet-18 model and report accuracy of our coresets in Table 2.Similar to other results, our method achieves better performance than randomly selected coresets for higher pruning rates and is competitive at lower pruning ratios. This is attributed to the difficulty of calculating concept similarity in the representation space of the CLIP model which is potentially unaware of the terminology used in the medical domain. While replacing CLIP with a VLM that is trained on medical domain can boost the performance of our method, our results highlight that even without access to such a model our approach is able to find better coresets than random subsets.

Our results on these two tasks highlight the versatility of our method for coreset selection, which is able to find coresets without requiring training the the downstream models on the entire dataset even once.

## C.3 Transferability of coresets

In this section, we present the performance of three additional downstream model architectures after training on the coresets found by our approach. Similar to the results in Table 4, our results in Table 14 show that coresets found by our approach in both standard and label-free setting achieve performance better than the random subsets on these three architectures as well. We note that the performance of the ViT model is worse than the performance of other model architectures with and without pruning (ViT-B-16 achieves an accuracy of $\approx 62\%$ compared to $\approx 78\%$ with ResNet-50 on full Imagenet dataset) due to the ViT models being data hungry in nature (Dosovitskiy et al., 2020) and the fact that we used standard SGD-based training (similar to that used for training other models in the paper). We believe that using other training methods for ViTs as suggested in (Touvron et al., 2021) could produce better performing ViT models.

Nonetheless, better performance than random subsets across a variety of downstream model architectures highlights the effectiveness of our approach at finding coresets without the knowledge of the architecture or training dynamics of the downstream models.

## C.4 Algorithm for stratified sampling using CCS (Zheng et al., 2022)

Here we present the algorithm for sampling the training examples to form the coreset based on the coverage-based selection methodology proposed by (Zheng et al., 2022). A crucial component of the algorithm is the cutoff rate $\beta$ which controls how many challenging samples should be removed from consideration when selecting the coreset. This is done to eliminate misclassified samples from the dataset since they can hurt the performance of the model trained on coreset, especially at high pruning rates. Previous works (Zheng et al., 2022; 2024) ablate the values of this cutoff ratio by training the downstream model on a range of values. In our work, we used the values proposed by the previous works and found that they work well for our score as well. In the following section, we present an ablation study for $\beta$ on CIFAR-100. The cutoff rates $\beta$ for different pruning rates $\alpha$ are as follows $(\alpha, \beta)$. For CIFAR-10: (30%, 0), (50%, 0), (70%, 10%), (90%, 30%), for CIFAR-100: (30%, 10%), (50%, 20%), (70%, 20%), (90%, 50%), for Imagenet: (30%, 0), (50%, 10%), (70%, 20%), (90%, 30%). We used CCS for label-free CS as well and the cutoff rates used were for CIFAR-10: (30%, 0), (50%, 0), (70%, 20%), (90%, 40%), for CIFAR-100: (30%, 0), (50%, 20%), (70%, 40%), (90%, 50%), for Imagenet: (30%, 0), (50%, 10%), (70%, 20%), (90%, 30%).

Table 14: Superior performance of downstream models with different architectures trained on our coresets for Imagenet compared to Random for both standard (Ours) and label-free (Ours-LF) CS highlights transferability of our coresets.

| Model Architecture | Method | Pruning Rates | | | |
|---|---|---|---|---|---|
| | | 30% | 50% | 70% | 90% |
| MobileNet | Random | $62.68_{\pm 0.09}$ | $62.49_{\pm 0.15}$ | $61.53_{\pm 0.29}$ | $53.80_{\pm 0.17}$ |
| | Ours-LF | $61.60_{\pm 0.18}$ | $61.97_{\pm 0.08}$ | $61.73_{\pm 0.25}$ | $54.39_{\pm 0.66}$ |
| | Ours | $61.36_{\pm 0.15}$ | $62.32_{\pm 0.22}$ | $62.46_{\pm 0.26}$ | $55.57_{\pm 0.13}$ |
| RN-34 | Random | $73.37_{\pm 0.08}$ | $71.71_{\pm 0.10}$ | $67.85_{\pm 0.04}$ | $51.29_{\pm 0.20}$ |
| | Ours-LF | $73.61_{\pm 0.08}$ | $71.99_{\pm 0.05}$ | $68.42_{\pm 0.21}$ | $53.52_{\pm 0.06}$ |
| | Ours | $73.39_{\pm 0.12}$ | $72.34_{\pm 0.13}$ | $69.44_{\pm 0.17}$ | $55.92_{\pm 0.02}$ |
| ViT-B-16 | Random | $59.09_{\pm 0.49}$ | $51.65_{\pm 0.46}$ | $40.19_{\pm 0.13}$ | $22.13_{\pm 0.16}$ |
| | Ours-LF | $57.50_{\pm 0.69}$ | $49.81_{\pm 0.89}$ | $40.33_{\pm 0.86}$ | $23.08_{\pm 0.27}$ |
| | Ours | $57.62_{\pm 0.56}$ | $52.67_{\pm 0.49}$ | $42.15_{\pm 0.81}$ | $24.43_{\pm 0.28}$ |

Table 15: Ablation of cutoff rate $\beta$ in CCS (Zheng et al., 2022) on CIFAR-100.

| Pruning Rate | $\beta = 0$ | $\beta = 0.1$ | $\beta = 0.2$ | $\beta = 0.3$ | $\beta = 0.5$ | $\beta = 0.7$ |
|---|---|---|---|---|---|---|
| 90% | 36.04 | 44.94 | 48.05 | 48.16 | **52.47** | 48.22 |
| 70% | 61.61 | 64.72 | **66.71** | 65.99 | 65.45 | x |
| 30% | 74.71 | **76.49** | 75.73 | x | x | x |

### C.4.1 Ablation on cutoff rate $\beta$

To evaluate the effect of $\beta$, we conduct an ablation study on CIFAR-100 using three pruning ratios: 30%, 70%, and 90%. For all pruning ratios, we observe that the accuracy of the model trained on the coreset initially increases and then decreases as $\beta$ increases. The initial improvement in accuracy suggests that removing a small number of very difficult samples can be beneficial, enabling the model to generalize better. However, removing too many hard samples reduces dataset coverage, which leads to performance degradation. Accordingly, for higher pruning ratios, we use larger values of $\beta$—in some cases setting it as high as $\beta = 50\%$. On the other hand, for lower pruning ratios, it is essential to retain hard samples to achieve high performance; thus, we use smaller values of $\beta$, typically setting it to 0% or 10%. These trends, summarized in Table 15, are consistent with findings from the CCS paper and other works on coreset selection (Sorscher et al., 2022; Paul et al., 2021). (In Table 15, entries marked with "x" indicate cases where $\beta \geq$ pruning rate, and are omitted because they would remove more samples than permitted by the target pruning ratio.)

### C.4.2 Ablation on number of bins $b$

To evaluate the effect of $b$, we tested five different values using a model trained on CIFAR-100 with a pruning ratio of 90%—this choice is motivated by Fig. 6(a) of CCS (Zheng et al., 2022), which showed that the value of $b$ has minimal impact at lower pruning ratios. Our results, summarized in Table 16, indicate that changing the bin size $b$ has only a small influence on the accuracy of the model trained on the coreset, consistent with observations in (Zheng et al., 2022). We observe that setting $b$ too small or too large can lead to slight

Table 16: Ablation of number of bins $b$ in CCS (Zheng et al., 2022)[1].

| Pruning Rate | $b = 25$ | $b = 50$ | $b = 75$ | $b = 100$ | $b = 150$ |
|---|---|---|---|---|---|
| 90% | 50.59 | **51.85** | 51.54 | 50.70 | 50.47 |

---

**Algorithm 2** Generate the Concept Bottleneck

---

**Require:** Class names $\mathcal{Y}$, concepts-per-class $k$, VLM, text encoder $T_{\text{enc}}$
**Ensure:** Concept list $C$, concept embedding matrix $E_C \in \mathbb{R}^{N_C \times d}$, per-class bottleneck concepts
  `bottleneck_concepts`
 1: {Step 1: Get raw concepts per class from VLM (class-name only, not per-image)}
 2: `raw_concepts` ← {} {map: class → list of strings}
 3: **for all** $y \in \mathcal{Y}$ **do**
 4:   prompt ← BuildPromptForDistinctAttributes($y$)
 5:   response ← VLM(`prompt`)
 6:   concepts$_y$ ← ParseAndCleanVLMResponse(`response`)
 7:   `raw_concepts`[$y$] ← concepts$_y$
 8: **end for**
 9: {Step 2: Select $k$ discriminative concepts per class}
10: `bottleneck_concepts` ← {} {map: class → list of length $k$}
11: **for all** $y \in \mathcal{Y}$ **do**
12:   disc ← SelectDiscriminative(`raw_concepts`[$y$], `raw_concepts`, `target_class` $= y$)
13:   `bottleneck_concepts`[$y$] ← [$y$] $\|$ TopK(`disc`, $k - 1$)
14: **end for**
15: {Step 3: Embed all concepts to form $E_C$ (rows are text embeddings of concepts)}
16: $C$ ← Flatten(`bottleneck_concepts` over all classes) {$|C| = N_C$}
17: $E_C \leftarrow \begin{bmatrix} T_{\text{enc}}(c_1)^\top \\ \vdots \\ T_{\text{enc}}(c_{N_C})^\top \end{bmatrix}$ {$E_C \in \mathbb{R}^{N_C \times d}$}
18: Return $(C, E_C, \text{bottleneck\_concepts})$

---

degradation in performance. However, we found that choosing $b \approx 50$ consistently yields high-performing coresets; this is the value we adopt throughout our work. (The results in Table 16 are averaged over three independent runs.)

## C.5 Additional experimental details

For generating the importance score we pre-compute the concept similarity scores for the entire dataset and then train the concept-bottleneck layer (in block 2 of Fig. 2) for 100 epochs across all experiments. This training only requires 800 seconds for Imagenet which is significantly more efficient than training the ResNet-34 model on Imagenet (requires roughly 8 hours on two A-100 GPUs). The accuracies of the models trained on the entire training set are 95.44% and 78.74% for ResNet(RN)-18 on CIFAR-10/100 and 72.4% for RN-18, 75% for RN-34, and 78.4% for RN-50 on Imagenet.

After the coresets are selected, we use the setting and code from (Zheng et al., 2022) for training a ResNet-18 model for 40000 iterations with a batch size of 256 on the coresets for all pruning ratios for CIFAR-10/CIFAR-100. For Imagenet, we train various models for 100 epochs on the coresets identified by our method using the training code based on FFCV (Leclerc et al., 2023).

The performance of the label-free CS is dependent on the quality of the pseudo-labels. Compared to the clustering-based approach used by ELFS (Zheng et al., 2024), our approach of using the zero-shot classification ability of CLIP models yields significantly better pseudo-label quality along with being simpler and more efficient to compute. Specifically, for CIFAR-10/100, pseudo-labels of the training set are computed using the CLIP L-14 model trained on the DataComp-1B dataset (Ilharco et al., 2021) yields an accuracy of 98.52% and 87.28% whereas for Imagenet it achieves an accuracy of 79.47% which are better than the best pseudo-label accuracy obtained by the clustering approach in ELFS (92.5% and 66.3% on CIFAR-10/100 and 58.8% on Imagenet).

For training the concept bottleneck layer we minimized the cross entropy loss using SGD with a learning rate of 1E-3, momentum of 0.9 and a weight decay of 5E-4 for 100 epochs.

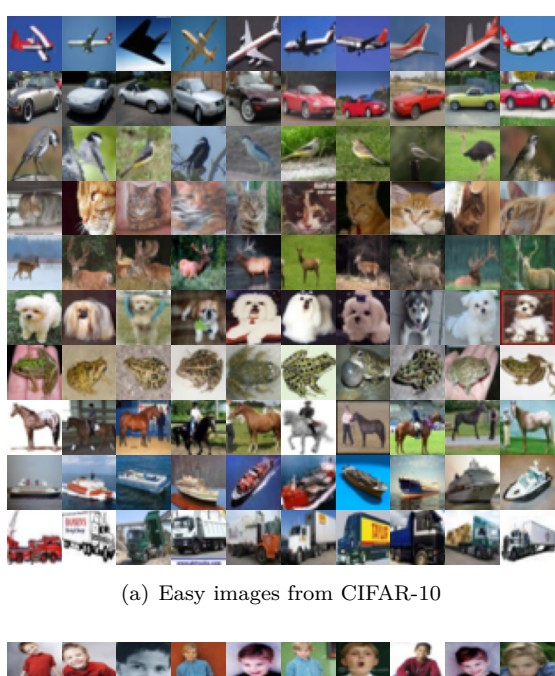

(a) Easy images from CIFAR-10

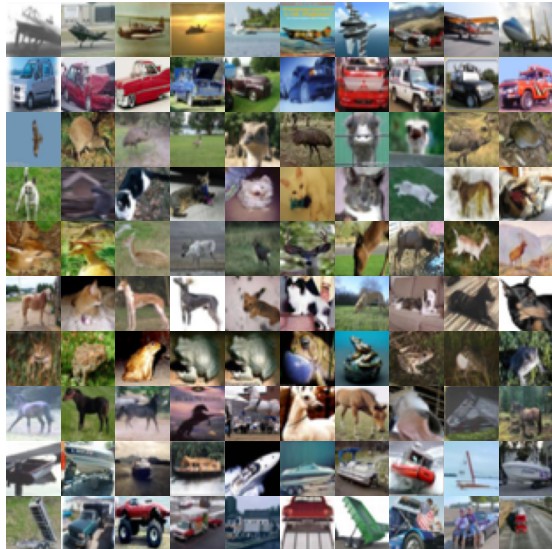

(b) Challenging images from CIFAR-10

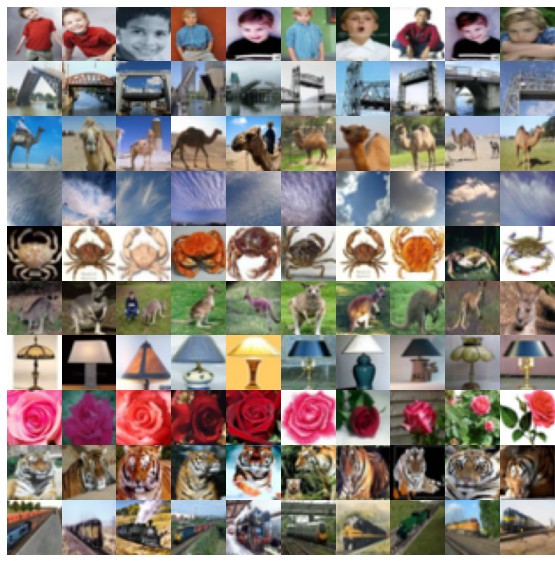

(c) Easy images from CIFAR-100

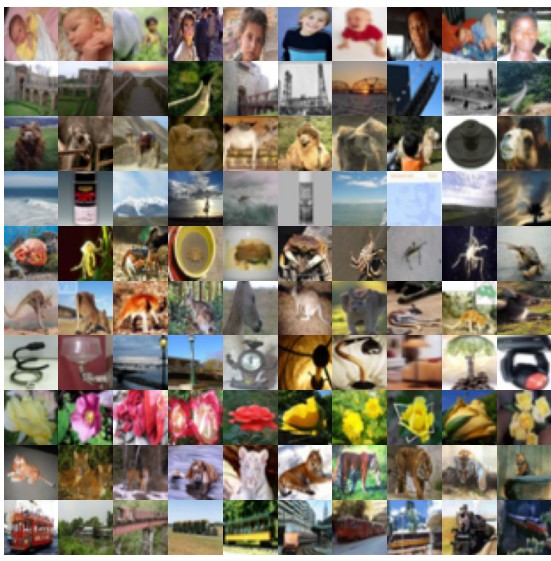

(d) Challenging images from CIFAR-100

Figure 4: Class-wise easy and challenging images for the 10 classes (`airplane, car, bird, cat, deer, dog, frog, horse, ship, truck`) in CIFAR-10 and for a subset of 10 classes (`boy, bridge, camel, cloud, crab, kangaroo, lamp, rose, tiger, train`) from CIFAR-100. Similar to the results in Fig. 3, easy images (a,c) are more canonical images associated with the class labels whereas challenging images (b,d) are images that are confused between two or more classes in the dataset.

---

**Algorithm 3** Concept-Based AUM Scoring

---

**Require:** Dataset $\mathcal{D} = \{(x_i, y_i)\}_{i=1}^n$, visual encoder $V_{\text{enc}}$, concept matrix $E_C \in \mathbb{R}^{N_C \times d}$, number of epochs $T$, optimizer settings

**Ensure:** Per-sample AUM scores and learned weights $W$

 1: Initialize $W \in \mathbb{R}^{N \times N_C}$
 2: Initialize $\texttt{sum\_margin}[i] \leftarrow 0 \ \forall i \in \{1, \ldots, n\}$
 3: **for** $t \leftarrow 1 \cdots T$ **do**
 4:     **for** minibatch $\mathcal{B} = \{(x_i, y_i, \text{idx}_i)\}$ from $\mathcal{D}$ **do**
 5:         {Concept similarity features (Eq. 2)}
 6:         $V \leftarrow V_{\text{enc}}(x_i) \ \{V \in \mathbb{R}^{|\mathcal{B}| \times d}\}$
 7:         $G \leftarrow V E_C^\top \ \{G \in \mathbb{R}^{|\mathcal{B}| \times N_C}\}$
 8:         {Linear predictor}
 9:         $\text{logits} \leftarrow G W^\top \ \{\in \mathbb{R}^{|\mathcal{B}| \times N}\}$
10:         {Cross-entropy loss (Eq. 3)}
11:         $\mathcal{L} \leftarrow \text{CE}(\text{logits}, y_i)$
12:         $W \leftarrow \text{OptimizerStep}(W, \mathcal{L})$
13:         {Margin accumulation for AUM (Eq. 4)}
14:         $h_t(x) \leftarrow \text{Softmax}(\text{logits})$
15:         **for all** $(s, y, \text{idx}) \in \mathcal{B}$ **do**
16:             $m \leftarrow s_y - \max_{y' \neq y} s_{y'}$
17:             $\texttt{sum\_margin}[\text{idx}] \leftarrow \texttt{sum\_margin}[\text{idx}] + m$
18:         **end for**
19:     **end for**
20: **end for**
21: $\text{AUM}_i \leftarrow \texttt{sum\_margin}[i]/T \ \forall i$
22: Return $\{\text{AUM}_i\}_{i=1}^n$, $W$

---

### C.6 Concept-based CS for tasks beyond image recognition

To validate the effectiveness of our concept-based coreset selection method beyond image recognition tasks, we apply our concept-based CS approach to the task of audio recognition.

We use the UrbanSound8k dataset (Arnault et al., 2020) for our experiments. The dataset contains 8732 labeled sound excerpts in .wav format each less than 4 seconds of urban sounds from 10 classes: `air_conditioner`, `car_horn`, `children_playing`, `dog_bark`, `drilling`, `engine_idling`, `gun_shot`, `jackhammer`, `siren`, and `street_music`. The classes are drawn from the urban sound taxonomy.

According to our approach we first use a language model, in this case GPT-5 to extract concepts for the 10 sound classes, using the following prompt, *"What are the distinct features that distinguish sound class name from other sounds. Give the output separated by commas in a line.".* We get $5 - 10$ distinctive concepts for every sound, for instance for sound class `children_playing`, we get the following concepts, *"youthful giggles", "running footsteps", "group chatter", "excited yelling", "playful screams", "playground noise".* We finally select $k = 5$ discriminative concepts from this list.

To test coreset performance, we use the CAM++ model (Wang et al., 2023) and report accuracy scores for our coresets in Table 17. When compared against randomly selected coresets for the various pruning ratios, coresets selected via our concept-based approach achieve better performance at various pruning rates.

---

**Algorithm 4** Coverage-centric Coreset Selection (CCS) (Zheng et al., 2022)

---

**Input**: Dataset with difficulty scores: $\mathbb{D} = \{(x, y, s)\}_{i=1}^n$, pruning ratio: $\alpha$, cutoff rate: $\beta$, number of bins: $b$.
**Output**: Coreset: $\mathcal{S}$

  # Prune hardest examples
  $\mathbb{D}' \leftarrow \mathbb{D} \setminus \{\lfloor n \times \beta \rfloor \text{ hardest examples}\}$
  $A_1, A_2, \cdots, A_b \leftarrow$ Split scores in $\mathbb{D}'$ into $b$ bins.
  $\mathcal{B} \leftarrow \{B_i : B_i \text{ consists of samples with scores in }.$
  $A_i \ for \ i = 1, \cdots, b\}$
  # Define the size of the coreset
  $m \leftarrow n \times \alpha.$
  **while** $\mathcal{B} \neq \varnothing$ **do**
    # Select the bin with the fewest examples
    $B_{min} \leftarrow \arg\min_{B \in \mathcal{B}} |B|.$
    # Compute the budgets for this bin
    $m_B \leftarrow \min\{|B_{min}|, \lfloor \frac{m}{|\mathcal{B}|} \rfloor\}.$
    $\mathcal{S}_B \leftarrow$ randomly sample $m_B$ samples from $B_{min}.$
    $\mathcal{C} \leftarrow \mathcal{C} \bigcup \mathcal{S}_B.$
    $\mathcal{B} \leftarrow \mathcal{B} \setminus \{B_{min}\}.$
    $m \leftarrow m - m_B.$
  **end while**
  return $\mathcal{C}.$

---

Table 17: Performance of concept-based coreset selection for **audio classification** task on UrbanSound8k dataset. Coresets selected by our approach results in superior performance compared to Random for standard CS at high pruning rates.

| Method | Pruning Rates | | | |
|---|---|---|---|---|
| | 30% | 50% | 70% | 90% |
| **Random** | $0.834_{\pm 0.00}$ | $0.782_{\pm 0.027}$ | $0.755_{\pm 0.023}$ | $0.746_{\pm 0.02}$ |
| **Ours** | $\mathbf{0.842}_{\pm 0.005}$ | $\mathbf{0.829}_{\pm 0.010}$ | $\mathbf{0.819}_{\pm 0.012}$ | $\mathbf{0.786}_{\pm 0.008}$ |

