# OpenReview forum: "Coreset Selection via LLM-based Concept Bottlenecks"
_TMLR — Rejected by TMLR_

### Review · Reviewer_qyBa · 2025-12-04

**Summary Of Contributions:**

This paper proposes a new coreset selection strategy based on concept bottlenecks derived from LLM and VLM. Instead of replying on training dynamics of downstream networks, the proposed method computes per-example difficulty by training a linear classifier on CLIP features projected through an LLaVA-generated concept bottleneck, then applying AUM and CCS for sampling. Experiments on CIFAR, ImageNet, AffectNet, and BloodMNIST show competitive accuracy to training-dynamics methods.

**Audience:**

Yes

**Audience Explanation:**

Although I am not expert in this domain, coreset selection and data pruning seem crucial to efficient large-scale training, and there is active interest in moving beyond brute-force scaling.

**Claims And Evidence:**

Yes

**Claims Explanation:**

The main empirical claims seem well supported.The paper demonstrates concpet-based coresets outperform arndom subsets across pruning rates and are competitive on three vision benchmarks. The efficiency claim is backed by wall-clock measurements.

**Requested Changes:**

- It seems CLIP model is a prerequisite for the proposed method in the label-free setting. I think this may be a limitation of the method.
- Authors should discuss comparisons with other model-agnostic data selection methods that do not require training the downstream model to help position this method.
- I suggest comparing CCS using the method's score vs CCS using simple CLIP-based scores to disentangle the concept-based score from CCS.
- Optional but I would suggest improving clarity in Section 4 by adding small pseudo-code blocks for the overall pipeline to help readers follow the flow.

---

> ### Author Response · Authors · 2025-12-29
> **Author response**
>
> We thank the reviewer for their review and questions. We provided answers and clarifications to their questions below.
>
> > It seems CLIP model is a prerequisite for the proposed method in the label-free setting. I think this may be a limitation of the method.
>
> Similar to how our work relies on the zero-shot prediction capability of CLIP-style models for pseudo labels for the label-free CS, SOTA approaches such as ELFS  (Zheng et al., 2024) have also utilized a clustering based approach (in the embedding space of a pre-trained DINO and SWAV model) for obtaining pseudo-labels. **Thus, this is a common approach for label-free CS in the literature and we believe it is not necessarily a limitation of our approach.**  We also emphasize that while we used CLIP models in our work, our method for label-free CS is compatible with any approach that can provide pseudo-labels (in the form of class names).
>
> > Authors should discuss comparisons with other model-agnostic data selection methods that do not require training the downstream model to help position this method.
>
> In our work we compared against several training dynamics based methods since they achieve SOTA results. Moreover, previous works such as DeepCore (Guo et. al 2022) and Coleman et al 2019, showed that training dynamics-free methods such as K-center greedy (Sener & Savarese, 2017) or Herding (Welling et al 2009) do not outperform the random subset baseline. Specifically, results in Table 1 and Table 2 of DeepCore (Guo et. al 2022) show that these methods underperform random subsets on CIFAR-10 and Imagenet-1K for the problem of standard coreset selection (where we have access to labels). Thus we omitted comparison to these methods in our Table 1. However, we can add these results to Table 1 to emphasize the sizable improvement our approach gets over other model-agnostic data selection methods.
>
> For label-free methods, we compared against prototypicality (Sorscher et. al) in our Table 3, where we see that this method underperforms the random subset and our approach provides a significant improvement, comparable to SOTA. (The result for the prototypicality approach was borrowed from ELFS (Zhang et. al 2024) Table 1 and Table 8.)
>
> New Reference:
>
> Welling, M.: Herding dynamical weights to learn. In: Proceedings of the 26th Annual International Conference on Machine Learning. pp. 11211128 (2009)
>
> > I suggest comparing CCS using the method's score vs CCS using simple CLIP-based scores to disentangle the concept-based score from CCS.
>
> **Below we present an experiment of comparing the performance of our concept-based score against using a clip-based score.** For computing the clip-based score we measure the similarity of the visual features of an image to the text captions of the form ``a photo of a {jth class name}’’ for all classes in the dataset. Then we use the score of the class with the maximum similarity as the clip score. For both scores, we used CCS to construct the coreset.
>
> We used CIFAR-100 with a 90% pruning rate similar to our ablation experiments in Sec 5.4. As can be seen in the result below, using just CLIP-based score leads to performance comparable to random subset selection. In contrast our method leads to a significant boost in the accuracy of the models trained on the coreset.
>
> | Method     | Accuracy  |
> |------------|--------|
> | Random     | 44.76 ± 1.58 |
> | CLIP-score | 42.58 ±  1.84|
> | Ours       | **51.85** ± 0.29 |
>
>
> > Optional but I would suggest improving clarity in Section 4 by adding small pseudo-code blocks for the overall pipeline to help readers follow the flow.
>
> **We have added three new algorithms to the paper to illustrate the flow of our proposed method.** The main algorithm is summarized in Algorithm 1. Algorithm 2 describes the process of generating concepts for the concept bottleneck layer, while Algorithm 3 summarizes the procedure for obtaining the concept-based score for each sample in the dataset. These are followed by the previously presented Algorithm 4 from CCS (Zhang et al., 2022), which generates the coreset via stratified sampling using the concept-based score.

---

### Review · Reviewer_Gygd · 2025-12-11

**Summary Of Contributions:**

The paper proposed an efficient algorithm for coreset selection in a training dataset. The method first trains a small linear concept bottleneck layer based on features derived from large language models, then computes a sample's difficulty score based on the alignment between that sample's visual features and the concept bottlenecks, and eventually, applies stratified sampling to generate the coreset. Empirical results on several image classification datasets demonstrate the effectiveness of the proposed approach over random baselines and comparable performance with SOTA methods.

**Audience:**

No

**Audience Explanation:**

The proposed method in some sense replaces the feature extraction in a model with embeddings from a pre-trained aligned VLM, which is why it only needs to train a single layer to predict how hard it is to align an input example in the pre-trained representation space. On the other hand, if we are leveraging the pre-trained VLM in the first place, why do we still need coreset selection to train other models for image classification?

Can the authors provide the image classification results by prompting existing VLMs used during feature extraction? That is a fairly important baseline missing in the paper.

In addition, can the authors provide additional experiment results beyond image classification, since the proposed method targets coreset selection, which is supposed to generalize to other tasks and domains?

**Claims And Evidence:**

No

**Claims Explanation:**

The empirical results in Tables 1 and 2 are not very competitive. The performance is only comparable to random pruning when the pruning rate is lower than 70%, and falls behind the SOTA baselines with clear margins. Although the SOTA baselines need training of the entire model, the size of the datasets compared in Table 1 is not too large to afford this additional cost.

And if the random baseline is strong, that means the datasets tested here are not challenging enough. Can the authors try other larger and harder datasets?

**Requested Changes:**

Both requests are critical to determining my evaluation of this paper:
1. Additional experiments on larger and more challenging datasets, tasks, and domains.
2. Additional baselines by directly prompting the VLMs used during the feature selection stage.

---

> ### Author Response · Authors · 2025-12-29
> **Author response (part 1 of 3)**
>
> We thank the reviewer for their review and questions. We provided answers and clarifications to their questions below.
>
> > The empirical results in Tables 1 and 2 are not very competitive.
>
> **We would like to clarify that the performance trend observed in our method with respect to pruning rates is consistent with trends reported in SOTA coreset selection literature.** As shown in Tables 1 and 3, where our method is compared against established baselines such as Forgetting and AUM (results taken from CCS Zheng et. al 2022) for standard coreset selection, and ELFS (Zheng et. al 2024) for label-free selection. Across these methods, we observe a common pattern: performance gains over random selection are marginal at moderate pruning rates (30%, 50%, 70%) but become more pronounced at higher pruning rates (90%).
>
> Additionally, our results in Table 2 on biomedical/affective computing tasks also show a similar pattern: while gains over random selection are smaller at moderate pruning levels, our method remains competitive with training-dynamics-based methods and outperforms random selection for higher pruning rates, despite not using training dynamics of the downstream model.
>
> This trend highlights an important characteristic of coreset selection: at moderate pruning levels, many samples are retained, reducing the performance gap across methods. In contrast, at higher pruning rates, careful selection becomes critical, which is where our method demonstrates its strength, achieving strong results without needing to train the downstream model on the entire data, offering up to 8× computational savings compared to methods that rely on training dynamics.
>
> For instance, training a model like ResNet-34 on ImageNet-1K to compute training dynamics for coreset selection takes approximately 8 hours. This cost has to be paid for every new model architecture since the coresets of these methods are architecture dependent. In contrast, our approach selects a high-quality coreset in under 30 minutes without requiring full model training or knowledge of the downstream architecture while achieving comparable or better performance.
>
> **We believe this efficiency–performance tradeoff makes our method highly practical for real-world scenarios where computational resources or labeled data are limited.**
>
>
> > If we are leveraging the pre-trained VLM in the first place, why do we still need coreset selection to train other models for image classification? Can the authors provide the image classification results by prompting existing VLMs used during feature extraction? Additional baselines by directly prompting the VLMs used during the feature selection stage.
>
>
> **While pre-trained Vision–Language Models (VLMs) such as CLIP provide strong zero-shot capabilities, they are not designed to fully replace supervised models for diverse downstream domains.** Our method leverages the representations from such VLMs to perform CS, for efficiently training smaller, task-specific models. This is particularly important because zero-shot VLMs, despite their generality, often underperform in specialized or domain-shifted settings.
>
> **To address the reviewer’s suggestion, we report below the zero-shot classification results obtained by directly prompting the CLIP ViT-B/32 model** used during feature extraction along with performance of our method with percentage of pruned data (sufficient to surpass zero-shot results of CLIP).
>
> | Dataset     | Metric        | CLIP ViT-B/32 | Ours CS (% Data Pruned) |
> |--------------|----------------|----------------|-------------------------|
> | CIFAR-10     | Accuracy (%)   | 88.78     | **91.80 (70%)**         |
> | CIFAR-100    | Accuracy (%)   | 61.69      |  **66.53 (70%)**         |
> | ImageNet     | Accuracy (%)   | 58.05     | **69.44 (70%)**         |
> | BloodMNIST   | Accuracy (%)   | 18.27     | **87.44 (90%)**         |
> | AffectNet    | F1 Score       | 0.27       |  **0.45 (90%)**          |
>
> As shown, CLIP performs competitively on natural image datasets (CIFAR, ImageNet) but struggles on non-natural domains such as AffectNet and BloodMNIST. In contrast, our coresets leads to models that (Tables 1 and 2) achieve significantly better results by pruning 70% of the data for CIFAR/Imagenet and pruning 90% data for Affectnet/BloodMNIST, i.e., training using only 10% of the data. These results are significantly higher than the corresponding zero-shot CLIP results, underscoring that CLIP alone cannot achieve optimal task performance.
>
> Coreset selection thus remains essential for producing compact yet representative subsets for training models on diverse downstream tasks.

---

> ### Author Response · Authors · 2025-12-29
> **Author response (part 2 of 3)**
>
> > Can the authors provide additional experimental results beyond image classification?
>
>
> Since the primary focus of our work is CS for classification tasks, we used image classification datasets in our evaluation to remain consistent with several state-of-the-art works in this area (Coleman *et al.*, 2019; Zhang *et al.*, 2023, 2024). However, below we present evaluations on three new tasks: CS for audio classification, CS for multi-label classification, and CS for generative modeling.
>
> **Concept-based CS for sound classification**
>
> For this evaluation, we use the UrbanSound8K dataset (Arnault *et al.*, 2020). The dataset contains 8,732 labeled sound excerpts in `.wav` format, each less than 4 seconds long, from 10 classes: *air_conditioner, car_horn, children_playing, dog_bark, drilling, engine_idling, gun_shot, jackhammer, siren,* and *street_music*.
>
> Using our approach, we first use a language model—GPT-5—to extract concepts for the 10 sound classes using the following prompt:
> *“What are the distinct features that distinguish {sound class name} from other sounds? Give the output separated by commas in a line.”*
>
> We obtain 5–10 distinctive concepts for every sound class. For example, for *children_playing*, we get: *“youthful giggles”, “running footsteps”, “group chatter”, “excited yelling”, “playful screams”, “playground noise”.* We then select *k = 5* discriminative concepts from this list.
>
> To test coreset performance, we use the CAM++ model (Wang *et al.*, 2023) and report accuracy scores in the table below. Compared to randomly selected coresets at various pruning ratios, our concept-based coresets achieve higher accuracy across all pruning rates.
>
> | **Method** | **30%** | **50%** | **70%** | **90%** |
> |-------------|----------|----------|----------|----------|
> | **Random**  | 83.4 ± 0.7 | 78.2 ± 2.7 | 75.5 ± 2.3 | 74.6 ± 2.0 |
> | **Ours**    | **84.2 ± 0.5** | **82.9 ± 1.0** | **81.9 ± 1.2** | **78.6 ± 0.8** |
>
> **Concept-based CS for multi-label classification**
>
> For this evaluation, we use the COCO 2017 dataset. We compare our approach against existing CS methods based on training dynamics (Forgetting and Area Under Margin, or AUM) and a clustering-based baseline.
>
> For the clustering-based method, we extracted features from a pretrained ResNet-101 model and performed *k-means* clustering to identify centroids. The importance of each sample was defined as its cosine distance to the nearest centroid—samples with the highest distances were chosen for the coreset.
>
> Since Forgetting, AUM, and our method were originally designed for single-label tasks, we adapted them as follows:
>
> - **Forgetting:** In single-label settings, forgetting is defined as the number of times a sample transitions from being predicted correctly to incorrectly during training. For the multi-label case, we define a sample as correct if at least one of its ground-truth labels is predicted correctly, and count transitions from correct to incorrect across epochs.
>
> - **Area Under Margin (AUM):** In single-label classification, AUM is computed using Eq. 4. In our multi-label adaptation, let $\mathcal{P}_x$ denote the set of positive (ground-truth) classes for a sample $x$, and $\mathcal{N}_x$ denote the set of negative classes (all others). The multi-label margin at epoch $t$ is defined as:
>
>   $$ \text{Margin}^{(t)}(x) = \min_{i \in \mathcal{P}x} h_i^{(t)}(x) - \max_{j \in \mathcal{N}_x} h_j^{(t)}(x)$$
>
>   The multi-label AUM is then the average margin over all $T$ epochs:
>
>   $$ AUM_{multi-label}(x)=\frac{1}{T}\sum_{t=1}^{T} \text{Margin}^{(t)}(x) $$
>
> - **Our Method:** We extracted 5 visual-semantic concepts per class using LLaVA for COCO. For each sample, we computed visual and concept similarity scores, trained a linear classifier on top, and used the modified AUM definition above to derive our concept-based importance score.
>
> We used stratified sampling to construct coresets for all methods. We applied a 90% pruning rate and performed linear probing on top of ResNet-101 features. Evaluation was done using mean Average Precision (mAP) on the COCO 2017 validation set.
>
> We observe that most existing methods perform slightly worse than the random baseline in this multi-label setting. However, our concept-based approach with the proposed AUM modification achieves the highest performance among all methods. This highlights the promise of concept-based coreset selection for multi-label tasks, although future work could further refine these scores.
>
> | **Method** | **mAP Score** |
> |-------------|---------------|
> | **Random** | 0.6695 |
> | **Forgetting** | 0.6492 |
> | **AUM** | 0.6580 |
> | **K-center** | 0.6238 |
> | **Ours** | **0.6720** |

---

> ### Author Response · Authors · 2025-12-29
> **Author response (part 3 of 3)**
>
> **Concept-based CS for generative modeling**
>
> Some recent works (e.g., Li *et al.* and Huang *et al.*) have begun exploring coreset selection (CS) for efficiently training generative models. Here, we use the coreset obtained through our proposed approach to train a class-conditional DiT-based (Diffusion Transformer) generative model on the ImageNet dataset.
>
> We assign an importance value to each sample within an ImageNet class using our concept-based scores and apply stratified sampling to select 50 images per class as part of the coreset. Since ImageNet contains 1,000 classes, the resulting coreset comprises 50,000 samples. Using this coreset, we train a DiT-B/2 model from scratch (keeping the pre-trained VAE fixed) for 300K steps.
>
> The table below reports the performance of the DiT model on standard generative metrics such as FID, Inception Score, and Precision/Recall. Our results show that the coreset derived from our approach leads to a substantially better FID after only 300K training steps compared to a randomly selected subset. Moreover, the FID achieved using our coreset is close to that obtained by training the DiT-B/2 model on the full ImageNet dataset for approximately 2 million steps (which requires about six days on two NVIDIA A100 GPUs). This demonstrates that our coreset—computed in just 30 minutes—can enable a generative model to reach competitive performance within a single day (on two A100 GPUs) of training.
>
>
> | **Metric**        | **Random** | **Our Coreset** |
> |--------------------|------------|----------------|
> | **Inception Score** (higher is better) | 73.53 | 118.91 |
> | **FID** (lower is better)            | 25.51 | 16.26 |
> | **Precision**       | 0.55  | 0.63 |
> | **Recall**          | 0.54  | 0.47 |
>
> For DiT training, we followed the implementation from the Fast-DiT repository, modifying only the dataset used for training. All hyperparameters were kept identical to the default configuration, except for the total number of training steps, which was set to 300K.
>
> For reference, the performance of the DiT model trained on the entire data using 2 million steps is as follows
>
> | **Metric**         | **Full Data (2M)** |
> |---------------------|--------------------|
> | **Inception Score**           | 130.26             |
> | **FID**                 | 11.00              |
> | **Precision**                       | 0.75               |
> | **Recall**                            | 0.64               |
>
> New References
>
> [a] Li, Yize, et al. “Pruning then Reweighting: Towards Data-Efficient Training of Diffusion Models.” ICASSP 2025 – IEEE International Conference on Acoustics, Speech and Signal Processing, IEEE, 2025.
>
> [b] Huang, Rui, et al. “Diffusion Dataset Condensation: Training Your Diffusion Model Faster with Less Data.” arXiv preprint arXiv:2507.05914 (2025).
>
> [c] Fast-DiT repository: https://github.com/chuanyangjin/fast-DiT/tree/main

---

> ### Comment · Reviewer_Gygd · 2026-01-14
>
> Thanks for the response. My concerns are mostly addressed.

---

> > ### Author Response · Authors · 2026-01-14
> > **Response by authors**
> >
> > We thank the reviewer for their response to our rebuttal and are glad that we could address their concerns.

---

### Review · Reviewer_5SpQ · 2025-12-24

**Summary Of Contributions:**

The paper proposes a Coreset Selection (CS) method that utilizes a lightweight model predicting the target label based on the similarity between the visual features and extracted concept embeddings of the target class. This method can also be extended to the unlabeled case, where a pseudo label can be selected from the extracted concepts. CS can be performed based on this (pseudo but also model agnostic) lightweight model, which offers up to 8x speedup compared to existing CS methods, while preserving comparable performance. Extensive empirical evaluations against training-dynamic based methods, and label-free methods are conducted on natural image datasets (CIFAR and Imagenet) and other datasets (BloodMNIST, AffectNet), showing the quantitative and qualitative effectiveness of the methods.

Strengths:

- The paper is very well written and most details are clear.
- The idea of performing CS on extracted concepts is nice and is effective.
- The experiments are quite extensive, covering several details about the effectiveness of the method.

Weaknesses:
- In general, this is a good paper. However, there are several details that make the paper incomplete:
- I’m concerned about the utility w.r.t different downstream models. I’d imagine that a user would select the best (known) model architecture and would perform CS on this model, but not on all models (as in Table 4, the performance is significantly different across different models). The only exception is when a SOTA model architecture is not the SOTA anymore, and the model-agnostic setting would be useful in this case. As a result, the advantage of model-agnostic becomes less crucial.
- The proposed Pseudo label is reasonable, but it’s not clear whether this work when the pseudo labels are more likely incorrect. I can see that the performance in this setting is just around random selection in Table 3 (especially for ImageNet), which perhaps makes random selection also a favorable approach. Do you have the experiments for AffectNet and BloodMNIST, where the pseudo labels could be more erroneous (although I’m not quite sure if this is true). In general, would it still work with less certain pseudo labels?
- In fact, the previous point is one of my main concerns for this work: what is the limitation of the proposed? The Random baseline works quite well in ImageNet and BloodMNIST (Tables 1-2), while AUM doesn’t work quite well. I also observe that the Prototypicality baseline performed very badly, compared to even Random (Table 3), which is different from what previously was reported in the corresponding paper.
-The experiment in Table 4/14 is nice, but one can also imagine a baseline selecting coreset using 1 model to train another model.
- I also think including the proxy baselines is useful for discussing the advantages of the method.
- “we select k discriminative concepts per class (concepts that are unique to a class) to form the concept bottleneck.” Perhaps an analysis of how many concepts that are unique would be useful. I would imagine in some datasets, the number of unique concepts would be limited. What is the dataset used in Table 7?
- The experiment on adaptive subset selection in Section 5.4 may not be well designed. I think adaptive CS changes based on model learning dynamic, but the proposed method (and Adaptive Random) is simply a random sample selection at each step, which makes its nature quite different from the other adaptive methods.

Minor Issues:

- grammatical errors: “focus of improving”, “on for this”, “learning. assumes an”.
- The statement “While interpretable, CBMs reliance on costly annotations and lower accuracy compared to end-to-end models limit their usage.” may not suitable given recent works on Concept Bottleneck Models, where these models match performance of the original models (see Sun et al. ICLR’25).

Sun et al. Concept bottleneck large language models. ICLR’25.

**Audience:**

Yes

**Audience Explanation:**

This problem of Coreset Selection is interesting and useful for efficient training of DNNs.

**Broader Impact Concerns:**

There's not much concern about the ethical implications of this work!

**Claims And Evidence:**

Yes

**Claims Explanation:**

The claims are being supported by a sound methodological design and experiments

**Requested Changes:**

As mentioned in the Weakness comments, the main concerns I have about the paper are the clarity and deeper analysis in the experimental section, as this paper's methodology is sound and simple. Please respond to my questions and concerns in that section.

---

> ### Author Response · Authors · 2025-12-29
> **Author response (part 1 of 2)**
>
> We thank the reviewer for their review and questions. We provided answers and clarifications to their questions below.
>
> > I’m concerned about the utility... model-agnostic becomes less crucial.
>
> We respectfully disagree with the reviewer’s assessment that identifying coresets in a model-agnostic manner is not crucial. This capability is, in fact, **highly valuable for two key reasons**.
>
> *First*, even when the SOTA model architecture for a dataset is known, computing the coreset using training-dynamics-based approaches still requires training the full model once to completion in order to capture its learning dynamics. This is computationally inefficient, particularly for large-scale datasets and deep architectures. Moreover, if a new architecture is later adopted, the entire coreset selection process must be repeated for that model, further increasing the computational cost.
>
> *Second*, our approach identifies the coreset in a fully data-centric and model-agnostic way. This allows the same coreset to be reused across multiple architectures, eliminating redundant computation and enabling broader applicability.
> As shown in Table 4, coresets identified by our method consistently lead to better performance than random subsets across diverse model architectures and pruning ratios. This demonstrates that a model-agnostic coreset not only improves efficiency but also generalizes well across architectures, making it a practically important property rather than a minor convenience.
>
> > The proposed Pseudo label is reasonable, ... less certain pseudo labels?
>
> We agree that the performance of label-free CS depends on the quality of pseudo labels. However, this dependency applies to both our method and SOTA training-dynamics-based approaches. For instance, in our Table 3, ELFS (Zheng et al., 2024) demonstrates that DINO-based pseudo labels outperform SwAV-based pseudo labels in the label-free setting.
> Similar to ELFS, our method consistently improves over Random subsets across all datasets. Moreover, our approach is computationally efficient and surpasses ELFS for certain pruning ratios on ImageNet, providing both an accuracy improvement and a computational advantage.
>
> **To further evaluate robustness under potentially noisy pseudo labels, we tested our label-free approach on BloodMNIST and AffectNet, generating pseudo labels with CLIP ViT-B-32.** The results (averaged over three runs) show that our method outperforms Random for most pruning ratios and achieves performance comparable to using true labels. While better pseudo labels can further improve results, our approach works effectively without any modification across biomedical and affective computing domains, highlighting its generality and practical utility.
>
> We thank the reviewer for this suggestion and will include these results in the revised paper.
>
> **Table:** Accuracy (%) across pruning ratios  on BloodMNIST dataset.
>
> | **Method**              | **30%** | **50%** | **70%** | **90%** |
> |--------------------------|:-------:|:-------:|:-------:|:-------:|
> | **Random**               | 94.99 ± 0.27 | 94.79 ± 0.18 | 91.71 ± 0.29 | 86.50 ± 0.66 |
> | **Ours (with labels)**   | 94.64 ± 0.60 | 94.06 ± 0.10 | 92.26 ± 0.19 | 87.44 ± 1.46 |
> | **Ours (label-free)**    | 95.28 ± 0.27 | 93.61 ± 0.35 | 91.86 ± 0.12 | 88.04 ± 1.01 |
>
>
> **Table:** F1 score across pruning ratios on Affectnet dataset.
>
> | **Method**              | **30%** | **50%** | **70%** | **90%** |
> |--------------------------|:-------:|:-------:|:-------:|:-------:|
> | **Random**               |0.607 ± 0.006 | 0.573 ± 0.025 | 0.517 ± 0.006 | 0.347 ± 0.055 |
> | **Ours (with labels)**   | 0.603 ± 0.006 | 0.577 ± 0.021 | 0.537 ± 0.015 | 0.450 ± 0.035 |
> | **Ours (label-free)**    | 0.596 ± 0.002 | 0.574 ± 0.002 | 0.513 ± 0.007 | 0.398 ± 0.022 |
>
>
> > In fact, the previous point ... previously was reported in the corresponding paper.
>
> This appears to be a misunderstanding. **For datasets such as CIFAR-10/100 and ImageNet (Table 1), we directly report the results from the prior work CCS (Zheng et al., 2023).** As shown there, AUM achieves significantly better performance than the Random baseline and remains state-of-the-art across most pruning ratios.For BloodMNIST (Table 2), AUM also performs better than the Forgetting baseline for the majority of pruning ratios.
>
> Similarly, **the results for the Prototypicality baseline in Table 3 are taken directly from the prior work on label-free CS, ELFS (Zheng et al., 2024)**, which demonstrated that approaches using training dynamics yield stronger coresets than both Random and Prototypicality. Our contribution builds upon these findings by showing that concept-based coreset selection using pseudo labels provides a computationally efficient alternative that outperforms both Random and Prototypicality, while offering comparable or better results without relying on any information about the downstream model architecture/training dynamics.

---

> ### Author Response · Authors · 2025-12-29
> **Author response (part 2 of 2)**
>
> > The experiment in Table 4/14 is nice, ...  advantages of the method.
>
> The idea of constructing a coreset by training one model to completion and then reusing that coreset to train another model has indeed been explored in approaches that rely on training dynamics to generate coresets (e.g., Zhang et al., 2023, 2024; Coleman et al., 2019). While such methods can yield good results, they require first training a model on the entire dataset, making them computationally expensive.
>
> **Our work directly addresses this limitation.** Unlike training-dynamics-based approaches, our method identifies the coreset without requiring knowledge of any downstream model and achieves an 8× efficiency improvement over methods that depend on full-model training. Moreover, as shown in Tables 1, 2, and 3, our approach attains comparable performance to coresets obtained through training-dynamics-based methods, while being significantly more efficient and scalable.
>
> > “we select k discriminative concepts per class (concepts that are unique to a class) ... dataset used in Table 7?
>
> The reviewer’s observation is correct — the number of unique or discriminative concepts available per class can indeed vary across datasets. This motivated us to include this experiment in our study to assess how the quality and distinctiveness of concepts across classes affect the concept-based score in our method. In Table 7, we used the CIFAR-100 dataset, and the results clearly show that having just five distinct concepts per class is sufficient to achieve strong performance.
>
> Furthermore, across all experiments in the paper—including those on ImageNet, AffectNet/BloodMNIST and UrbanSounds8K (results in our response to Q3 by reviewer Gygd), **we found that modern VLMs can readily generate around five meaningful and distinct concepts per class without extensive prompt engineering, demonstrating the practicality of our approach.**
>
> > The experiment on adaptive subset selection in ... methods.
>
> The reviewer’s intuition is correct that adaptive subset selection methods dynamically select the next subset of data for training based on the model’s current state. The primary motivation for including this experiment is to demonstrate that our method—originally designed for static coreset selection problems (where the coreset remains fixed during training) can be minimally modified to function effectively in adaptive subset selection scenarios as well. As shown in Table 13, our approach achieves competitive performance on this task, despite not being specifically optimized for adaptive selection.
>
> That said, **this experiment was included mainly to acknowledge and differentiate our work from prior research in adaptive coreset selection.** We are open to removing it or moving it to the appendix if the reviewer believes it would improve the clarity and focus of the paper.
>
> > Minor: 1. Grammatical errors 2. Fix statement about CBMs.
>
> We will fix the grammatical errors and the statement about CBMs in the revised manuscript and will cite the new paper provided by the reviewer.

---

> > ### Comment · Reviewer_5SpQ · 2026-01-31
> >
> > Thank you for the responses, and I'm very sorry for the late response (due to family matters). Nevertheless, I have read the responses and considered them in my final recommendations.

---

> > > ### Author Response · Authors · 2026-01-31
> > > **Response to reviewer**
> > >
> > > We thank the reviewer for their response. If there are any other concerns that could help improve the rating, we would be happy to address them.

---

### Decision · Action_Editor_SRbk · 2026-02-02

**Recommendation:** Reject

**Audience:**

No

**Audience Explanation:**

In its current form, the paper is unlikely to stimulate meaningful discussion within TMLR, as substantial improvements in both quality and soundness are required.

**Claims And Evidence:**

No

**Claims Explanation:**

The paper proposes a coreset selection strategy based on concept bottlenecks derived from large language models (LLMs) and vision–language models (VLMs). The authors have made a commendable effort on the experimental side, which has been positively acknowledged by all reviewers.

Despite the paper’s simplicity and clarity, the reviewers’ final recommendations reflect a mixed consensus. Importantly, several issues related to the quality and soundness of the current manuscript remain unresolved.

In particular, key claims are insufficiently supported:
1. The paper lacks adequate contextualization with respect to existing work on dataset distillation and condensation (see, for example, [1, 2, 3]).
2. The main claim of avoiding expensive pre-training does not accurately reflect the paper’s actual contribution. The proposed method fundamentally relies on pre-trained models as the basis of the algorithm. As such, it neither resolves nor circumvents the core issue of costly pre-training, and therefore does not meaningfully contribute to reducing this cost, contrary to what is claimed.
3. In addition to these concerns, and as also noted by reviewers 5SpQ and qyBa, it remains unclear what advantages the proposed method offers over a random baseline. The significance of the approach should be more clearly articulated and justified.

**References** \
[1] T. Wang, J. Y. Zhu, A. Torralba, A. and A. A. Efros. Dataset Distillation. 2018. \
[2] G. Cazenavette, T. Wang, A. Torralba, A. A. Efros, J. Y. Zhu. Dataset Distillation by Matching Training Trajectories. CVPR 2022. \
[3] B. Zhao, K. R. Mopuri, H. Bilen. Dataset Condensation with Gradient Matching. ICLR 2021.

**Resubmission Of Major Revision:**

The authors may consider submitting a major revision at a later time.